# Myriad Mapping of nanoscale minerals reveals calcium carbonate hemihydrate in forming nacre and coral biominerals

Connor A. Schmidt[1], Eric Tambutté [2], Alexander A. Venn [2], Zhaoyong Zou [3], Cristina Castillo Alvarez [4], Laurent S. Devriendt[4], Hans A. Bechtel [5], Cayla A. Stifler[1], Samantha Anglemyer [1], Carolyn P. Breit[1], Connor L. Foust[1], Andrii Hopanchuk[1], Connor N. Klaus[1], Isaac J. Kohler [1], Isabelle M. LeCloux[1], Jaiden Mezera [1], Madeline R. Patton [1], Annie Purisch[1], Virginia Quach[1], Jaden S. Sengkhammee[1], Tarak Sristy [1], Shreya Vattem[1], Evan J. Walch[1], Marie Albéric[6], Yael Politi[7], Peter Fratzl[8], Sylvie Tambutté [2] & Pupa U.P.A. Gilbert [1,4,9,10] ✉

Calcium carbonate ($CaCO_3$) is abundant on Earth, is a major component of marine biominerals and thus of sedimentary and metamorphic rocks and it plays a major role in the global carbon cycle by storing atmospheric $CO_2$ into solid biominerals. Six crystalline polymorphs of $CaCO_3$ are known−3 anhydrous: calcite, aragonite, vaterite, and 3 hydrated: ikaite ($CaCO_3 \cdot 6H_2O$), monohydrocalcite ($CaCO_3 \cdot 1H_2O$, MHC), and calcium carbonate hemihydrate ($CaCO_3 \cdot \frac{1}{2}H_2O$, CCHH). CCHH was recently discovered and characterized, but exclusively as a synthetic material, not as a naturally occurring mineral. Here, analyzing 200 million spectra with Myriad Mapping (MM) of nanoscale mineral phases, we find CCHH and MHC, along with amorphous precursors, on freshly deposited coral skeleton and nacre surfaces, but not on sea urchin spines. Thus, biomineralization pathways are more complex and diverse than previously understood, opening new questions on isotopes and climate. Crystalline precursors are more accessible than amorphous ones to other spectroscopies and diffraction, in natural and bio-inspired materials.

The marine calcium carbonate ($CaCO_3$) factory[1] is a major component of the global carbon cycle as it removes $CO_2$ from the atmosphere and generates solid minerals. The most significant fraction of marine carbonate minerals are the biominerals deposited by living organisms, such as mollusk shells[2–5], coral skeletons[6,7], coccoliths, and foraminifera tests[8]. These biominerals consistently outperform the sum of their components, inspiring strategies to design novel materials and metamaterials[9–15]. Ever since their proliferation in the Cambrian[16], marine carbonate biominerals have changed the carbon cycle, the seawater composition, and formed the majority of marine

[1]Department of Physics, University of Wisconsin, Madison, WI 53706, USA. [2]Department of Marine Biology, Centre Scientifique de Monaco, 98000 Monaco, Principality of Monaco. [3]State Key Laboratory of Advanced Technology for Materials Synthesis and Processing, Wuhan University of Technology, Wuhan 430070, China. [4]Chemical Sciences Division, Lawrence Berkeley National Laboratory, Berkeley, CA 94720, USA. [5]Advanced Light Source, Lawrence Berkeley National Laboratory, Berkeley, CA 94720, USA. [6]Sorbonne Université/CNRS, Laboratoire de chimie de la matière condensée, 75005 Paris, France. [7]B CUBE - Center for Molecular Bioengineering, Technische Universität Dresden, 01307 Dresden, Germany. [8]Max Planck Institute of Colloids and Interfaces, 14476 Potsdam, Germany. [9]Departments of Chemistry, Materials Science and Engineering, and Geoscience, University of Wisconsin, Madison, WI 53706, USA. [10]Previously publishing as Gelsomina De Stasio. ✉e-mail: pupa@physics.wisc.edu

sedimentary rocks[1]. Several marine carbonate biominerals were shown to form via precursor amorphous calcium carbonate phases[17–25], but biomineral formation mechanisms are hotly debated[26], especially because they determine the resilience of biominerals to climate change, including ocean warming and acidification[16]. Here we show that during biomineralization four distinct mineral phases are present, two amorphous and two crystalline, all four transiently present, localized on the surface of forming biominerals, whereas in the bulk of all biominerals the precursors crystallize into mature, stable mineral phases.

## Results

Below we describe how we observed two unknown $CaCO_3$ mineral phases using soft-x-ray spectroscopy at the nanoscale, how we identified them as CCHH and MHC, and therefore needed to develop a new method that allows to quantitatively display many more phases. Using the new method, we observed the spatial distribution of five distinct phases and their occurrence across diverse biominerals. We also provide corroborating evidence from a completely independent infrared method.

### Unknown $CaCO_3$ mineral phases

While analyzing amorphous precursor phases in freshly deposited biominerals using x-ray absorption spectromicroscopy[27–32], we noticed two mineral phases whose calcium L-edge spectra did not match any of the known precursor or mature phases.

The known mature phases were aragonite, calcite, or vaterite, the three anhydrous polymorphs of $CaCO_3$, and their amorphous

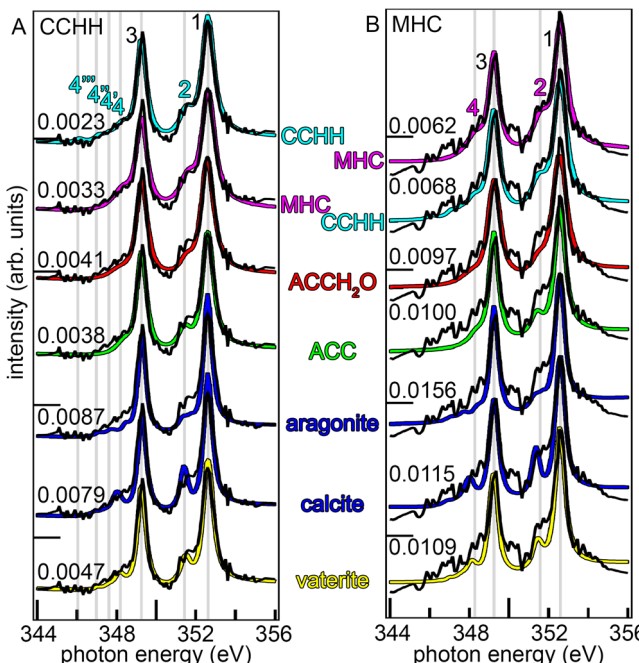

**Fig. 1 | Identifying the unknown mineral phases.** An unknown spectrum, plotted in black, extracted from a 62 nm single pixel at the surface of a coral skeleton, assigned 95% CCHH by analysis with all components. Reference spectra for all phases, amorphous and crystalline, from synthetic minerals are shown in colors and displaced vertically for clarity. Each reference spectrum is overlapped by the unknown single-pixel spectrum. The $χ^2$ value for each fit with 1 component is displayed to the left of each reference spectrum. Gray vertical lines indicate peak positions for the 7 peaks characteristic of CCHH. **B** An unknown spectrum from a 57 nm single pixel at the surface of a forming nacre tablet, assigned 82% MHC by analysis with all components. Again, $χ^2$ values of each fit are displayed on the left of each reference spectrum. Gray vertical lines indicate peak positions for the 4 MHC characteristic peaks.

precursors, including amorphous calcium carbonate hydrated or anhydrous ($ACCH_2O$ or ACC, respectively)[17–25]. The newly observed spectra did not match any of these previously observed phases, thus, we set out to identify them spectroscopically.

### Identification as CCHH and MHC

The spectra that best matched the unknown spectra were calcium carbonate hemihydrate ($CaCO_3·\frac{1}{2}H_2O$, CCHH) a new synthetic material recently discovered[33], and monohydrocalcite ($CaCO_3·1H_2O$, MHC)[33]. The spectral match was significantly better for these two phases than for any other amorphous or crystalline phases, as indicated by much lower $χ^2$ in Fig. 1.

The unknown spectrum in Fig. 1A best matches the CCHH reference spectrum, as indicated by the lowest $χ^2$ value of 0.0023. Notice the small, but distinct peaks at 346.1 eV and 351.5 eV indicated by gray vertical lines in Fig. 1A—and labeled peak and peak 2, which are characteristic of CCHH. Peak is not distinguishable from the noise in the unknown spectrum, but peak 2 is. Furthermore, the relative intensities of all peaks with respect to the main peaks 1 and 3 are characteristic of each phase.

The unknown spectrum in Fig. 1B best matches the MHC reference spectrum, with $χ^2 = 0.0062$. The MHC spectrum has broader, less resolved peaks 2 and 4. $ACCH_2O$ also has similarly broad peaks, but the peak intensities are very different for $ACCH_2O$ and MHC. Again, the single-pixel spectrum is noisy, but its overall lineshape and peak intensities are clearly distinguishable. This is why the $χ^2$ values obtained by fitting the unknown spectrum with either MHC or $ACCH_2O$ are very different: $χ^2 = 0.0062$ and $χ^2 = 0.0097$, respectively.

The CCHH and MHC characteristic peak positions are highlighted by gray vertical lines in Fig. 1, so the reader can observe the shifts in energy of relevant peaks in all other known spectra compared to CCHH and MHC. Also, the CCHH peak positions are indicated by cyan vertical lines in Supplementary Fig. 1C, D. The area under each peak (termed Amplitude here) and all peak intensities are also clearly different across the known spectra. In Supplementary Fig. 1C, D, peak positions and intensities can directly be compared across phases: aragonite has the lowest intensity peak 2, calcite has the highest, the others have intermediate intensities but their energy positions vary (Supplementary Fig. 1D). MHC has the highest peak 4 and aragonite has the lowest, the other phases have intermediate intensities (Supplementary Fig. 1C). The smaller crystalline peaks labeled 4', 4", 4'" in Fig. 1 and Supplementary Fig. 1 are too low in intensity to be distinguished from the noise in single-pixel spectra, but they stand out when averaging more than 20 pixels. Even though these small peaks can't be distinguished, the lineshape of the noisy spectrum in this region is distinct from the valley that exists in this region in ACC and $ACCH_2O$ spectra (Fig. 1).

### New method: Myriad Map(ping) (MM) of nanoscale mineral-phases

The identification of CCHH and MHC as potential crystalline precursors to aragonite and calcite during biominerals' formation, along with previously well-established $ACCH_2O$ and ACC, made it necessary to develop a new method to display the distribution of four metastable precursors and one stable crystalline phase, all in the same image. Previous methods such as component mapping[21–25] were limited to three components only, because computer graphics only allow 3-color mapping to be quantitative, either as RGB or CMY imaging. We therefore developed Myriad Mapping (MM) for nanoscale mineral-phases, where any number of phases could be displayed at once. The word Myriad highlights that this method displays quantitatively an unlimited number of phases. For human observation at a glance, a dozen phases can easily be distinguished. All MMs were acquired with microscale field of view and nanometer resolution, but MM can be used for quantitative imaging of any size, from intergalactic to atomic

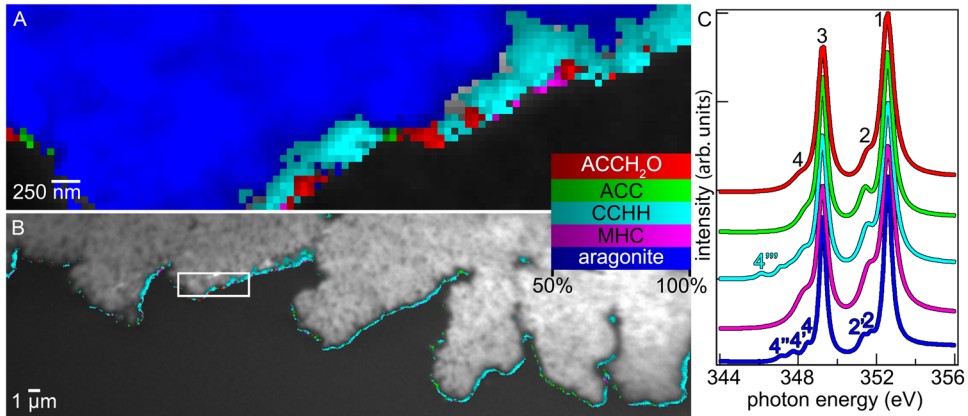

**Fig. 2 | CCHH on the surface of coral skeleton.** CCHH on the surface of a *Stylophora pistillata* coral skeleton. **A, B** Grayscale photoelectron image of a coral skeleton (top) with tissue and embedding material (bottom). The box in (**B**) indicates the region magnified in (**A**). In both panels, the colored pixels superimposed on the grayscale micrograph are carbonate Myriad Maps (MMs) of nanoscale mineral phases, displaying only pixels that contained 50% or more of each phase, color coded so red = ACCH₂O, green = ACC, cyan = CCHH, magenta = MHC, blue = aragonite, with brighter/darker colors corresponding to greater/lower concentration (see color legend). In (**B**), the aragonite blue pixels are not displayed so the morphology of the skeleton is visible. This area was analyzed in duplicate with consistent results. **C** Ca L-edge x-ray absorption spectra of 5 calcium carbonate phases, acquired from synthetic reference minerals, used for MMs and color-coded as in (**A**), (**B**). The spectra were displaced vertically for clarity. Color blind readers are referred to Supplementary Fig. 12 to see these same data in different colors.

scales. MM was achieved by assigning a color to each phase, displaying the concentration of that phase in each pixel as a duotone image, where black = 0% and bright color = 100% concentration of that phase, and every concentration in between has intermediate brightness. Then, only the pixels with concentration greater than 50% were selected from each duotone image and displayed as a Photoshop® layer. This operation was repeated for each phase, duotone images were stacked in multiple layers, and the result was a MM where the majority phase and its concentration were displayed in each pixel. Different phases or colors never overlapped in MMs because only one phase could ever have >50% concentration in each image pixel. Thus, information about mixed-phase pixels was lost, but the maps presented no ambiguity about the assignment of each pixel to a phase, and they enabled the display of as many phases as needed by the system under analysis. In Fig. 2 the MM of a coral skeleton surface is presented.

In all areas analyzed, the unknown-composition spectrum for each pixel was fit by a linear combination of carbonate phase spectra (Fig. 2C, Supplementary Table 1, Supplementary Fig. 1, Supplementary Data 1.xlsx) and the fit goodness was measured by the $\chi^2$ value. CCHH has only been identified synthetically, thus, all carbonate phase spectra presented in Fig. 1 come from synthetic mineral sources (see Supplementary Table 1). For comparison between the fits of synthetically and biogenically sourced carbonate phase spectra, see Supplementary Table 2. Five-component analysis was consistently significantly better than 3-component (p = 3.0 × 10⁻²⁰⁷ for CCHH pixels), whereas 6-components including vaterite did not significantly improve the analysis (p = 0.63 for vaterite pixels) (see Supplementary Table 3 and Supplementary Data 2.xlsx for detailed statistical analysis and comparisons). Thus, vaterite was not included in any MMs in this work.

Notice in Fig. 2C that the peaks labeled 1 and 3 were common to all carbonate phases, whereas peaks 2 and 4 were distinct for each phase. Peak 2 was a single peak for all phases except for aragonite where it split into two peaks (2, 2'). Peak 4 was a single peak in ACCH₂O, ACC, and MHC, it split into three peaks in aragonite (4, 4', 4'') and four peaks in CCHH (4, 4', 4'', 4'''). See Supplementary Figs. 1–3 for greater magnification on these peaks, examples of how peak fitted spectra and example residues, and Supplementary Table 4 for the precise amplitudes, positions, and widths of all peaks. The fit of unknown spectra with known component spectra included all peak parameters: position, amplitude, and width for each known phase, thus each fit took into account all peaks and their peak parameters.

## Spatial distribution of phases

The MM in Fig. 2 shows that CCHH and MHC spectra mapped in a characteristic pattern: they clustered in space, rather than being randomly distributed as single pixels interspersed with other amorphous or crystalline phases, and they were observed to accumulate preferentially at the surface of forming biominerals, not in the bulk, where the mature, stable, crystalline aragonite was found. Notice that abundant CCHH was identified all over the freshly deposited surface of coral skeletons (Fig. 2B). Pixels with 50% or more aragonite were space-filling, colored in blue, in the skeleton bulk (top left in Fig. 2A). A few blank pixels were visible near the skeleton surface in Fig. 2A, where the underlying grayscale image is visible. These were pixels with three or more mixed phases, all of which were <50%, thus, they contained CaCO₃ minerals but were not displayed. CCHH and ACC were the most abundant precursor phases found near the surface of the biomineral, forming a discontinuous layer with thickness between 0-800 nm of either or both phases. ACCH₂O, and MHC, appeared in a thinner, even more discontinuous layer at the outermost surface, less than 100 nm in thickness (Fig. 2B).

## Occurrence in diverse phyla

In addition to coral skeletons, CCHH and MHC were found in even greater abundances in mollusk shell nacre, but only in trace amounts in sea urchin spines. Representative MMs of the three biominerals are presented in Fig. 3.

These results were observed consistently across 86 areas, ranging in size between (20–65 μm)², of which 54 were in 8 coral skeletons: 3 *Stylophora pistillata* and 5 other coral species; 16 areas in nacre: 2 *Haliotis rufescens* shells; 16 areas in 5 sea urchin spines from 5 different animals: 4 *Strongylocentrotus purpuratus*, and 1 *Paracentrotus lividus*. All 86 areas were acquired in duplicate or triplicate, for a total of 180 separate acquisitions, containing more than 1.95 × 10⁸ spectra in total. The sample size, therefore, was orders of magnitude greater than necessary for statistical significance. A summary of the MM results for all spectra in all pixels is presented in Supplementary Table 5. The extensive results are presented in the file Supplementary Data 2.xlsx.

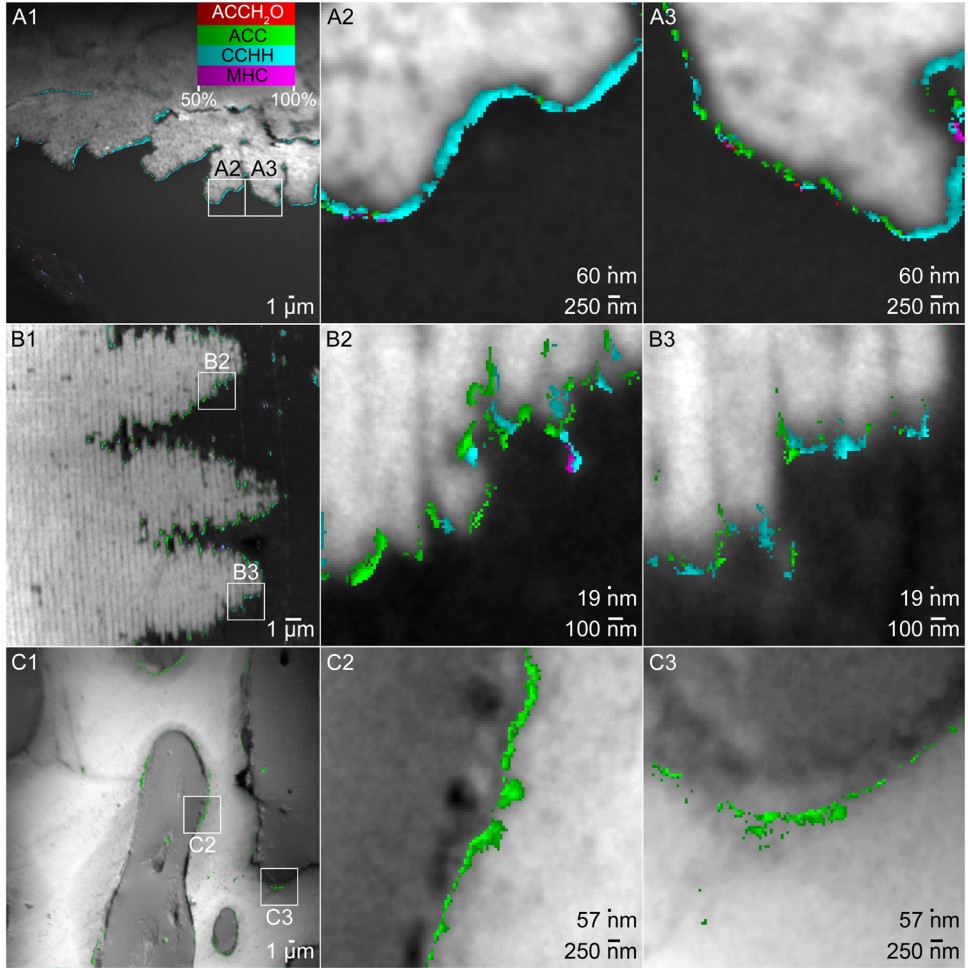

**Fig. 3 | CCHH on the surface of coral skeleton and nacre, not on sea urchin spine. A** Myriad Map (MM) of coral skeleton from the smooth cauliflower coral *Stylophora pistillata*. **B** MM of nacre from the California red abalone *Haliotis rufescens*. **C** MM of sea urchin spine from the California purple sea urchin *Strongylocentrotus purpuratus*. All biominerals are observed in cross-section. The boxes in panels 1 indicate the regions magnified in panels 2 and 3. For all biominerals, the background is a grayscale photoelectron image of the area; the superimposed colored pixels are MMs obtained using 5 component spectra (Fig. 2C,

Supplementary Fig. 1). Colored pixels in all panels represent pixels containing more than 50% of the indicated mineral phase, and are color coded according to the color legend in A1. For clarity, blue pixels of aragonite (for coral and nacre) and calcite (for sea urchin spine) are omitted from all panels. They are included in Supplementary Figs. 4–6, where all phases are displayed for all pixels in the first and second acquisition. All three areas were analyzed in duplicate with consistent results.

## Corroborating evidence from surface-sensitive infrared

In addition to x-ray absorption spectroscopy, we looked for transient crystalline precursors in forming biominerals using synchrotron infrared nanospectroscopy (SINS), which used Fourier transform infrared (FTIR) spectroscopy on two freshly deposited *Stylophora pistillata* coral skeletons. We first acquired reference standard spectra from synthetic mineral phases, then acquired spectra near the forming edge of the coral skeletons and compared the biogenic to the synthetic spectra to determine if transient precursor phases were observed in coral skeletons. Representative spectra are presented in Fig. 4, for forming and mature coral skeleton and synthetic minerals. We found that many of the $(20 \text{ nm})^3$ voxels at the surface of the skeleton yielded spectra with the $v_3$ peak split into 3 or 2 peaks. Such splitting is characteristic of and therefore identifies CCHH and MHC, respectively (Fig. 4A–C).

The frequency positions of $v_3$ peaks in coral skeleton surfaces were shifted with respect to synthetic CCHH and MHC minerals, but the peak splitting and relative positions of the 3 or 2 $v_3$ peaks remained constant and therefore enabled CCHH and MHC identification. These peaks and their shifts were highlighted by black vertical lines in Fig. 4A–C. In Fig. 4A, the SINS-FTIR spectrum acquired from the freshly

deposited coral surface I from a *Stylophora pistillata* coral skeleton (Supplementary Fig. 7A, B) is presented and compared to one of the synthetic CCHH spectra. In the coral surface I spectrum, the carbonate $v_3$ peak is split into 3 peaks labeled $v_3'$, $v_3''$, $v_3'''$, which are characteristic of CCHH. This split indicates that there are 3 C–O bond asymmetric stretching modes[34]. The $v_3$ peaks in coral surface I are rigidly shifted down in frequency relative to synthetic (syn) CCHH I by -50 cm⁻¹. In coral surface I, both peaks $v_3'$ and $v_3'''$ correspond to aragonite peaks, thus in this position there was a mixture of CCHH and aragonite. Another SINS-FTIR spectrum labeled coral surface II is presented in Fig. 4B, from the same coral skeleton, compared to a different synthetic CCHH spectrum, syn CCHH II, acquired from the same sample as syn CCHH I. The peak positions and splitting in the $v_3$ region vary significantly across the coral surface spectra, which may correspond to multiple levels of hydration. Similar variability was also observed across different grains of synthetic CCHH powder, as shown by a comparison of syn CCHH I and II spectra in A and B. The peak shift between syn CCHH II and coral surface II is not rigid, with each $v_3$ peak shifted by a slightly different amount, but still down in frequency by 40–60 cm⁻¹. The $v_3'$ peak is also shifted towards lower frequencies relative to aragonite, by -15 cm⁻¹. Finally, in Fig. 4C. another SINS-FTIR

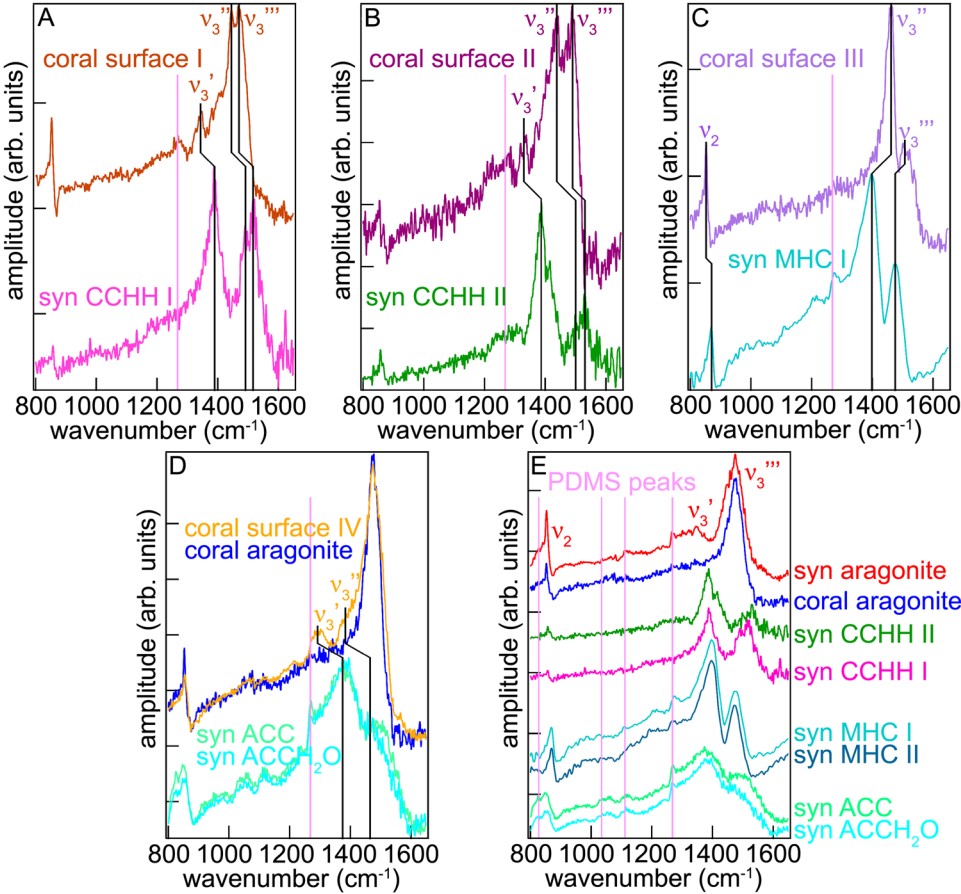

**Fig. 4 | Validation from infrared spectroscopy.** Synchrotron Infrared Nano Spectroscopy (SINS) Fourier Transform InfraRed (FTIR) spectra from coral skeletons and synthetic carbonates acquired with 20 nm resolution. The top spectra in (**A–D**) were acquired from forming coral surfaces, and the best matching spectra from their synthetic counterparts are shown at the bottom. SINS-FTIR spectra acquired from synthetic carbonates, amorphous, metastable crystalline, and stable aragonite are shown in E, compared with pure aragonite extracted from a coral skeleton. All polydimethylsiloxane (PDMS) contaminant peaks are indicated by pink vertical lines and described in more detail in the Methods section.

spectrum from a different coral sample (Supplementary Fig. 7C, D), coral surface III, compared to synthetic MHC. The lineshape of $v_3$ is very similar between the two spectra, with the coral shifted up in frequency by $35–65\,cm^{-1}$. These shifts in the $v_3$ peaks of biogenic CCHH and MHC and change in peak splitting could be due to multiple factors, including mixed signal from aragonite, different hydration levels relative to their synthetic counterparts, interactions with organic molecules trapped within the biomineral, or trace elements in the biominerals, any of which could potentially cause a change in the asymmetric stretching of the carbon-oxygen bonds in carbonate ions[33]. In coral skeletons, CCHH and MHC $v_3$ peak positions and intensities varied from voxel to voxel, but so did, to a smaller extent, synthetic CCHH and MHC, as shown in Fig. 4E.

In addition, we found amorphous precursors on the surface but not the bulk of coral skeletons. Figure 4D shows the orange coral surface IV (Supplementary Fig. 7A, B) spectrum, which is mostly aragonite, but with a slightly broader $v_3$ peak compared to the coral aragonite spectrum, acquired from a mature area of a different *Stylophora pistillata* coral skeleton (Supplementary Fig. 7C, D), which indicates greater structural disorder. In addition, this coral surface spectrum shows two broad peaks, indicated by vertical lines, which correspond exactly to those observed in synthetic $ACCH_2O$. The two peaks in coral are shifted, by the same amount ($\sim80\,cm^{-1}$), with respect to synthetic $ACCH_2O$. $ACCH_2O$, therefore, was detectable, even though SINS is less surface sensitive than PEEM (20 nm vs. 3 nm at the Ca L-edge[35]).

Figure 4E shows SINS spectra of synthetic carbonates, amorphous, metastable crystalline, and stable aragonite. The coral aragonite spectrum from D is also shown for comparison to the synthetic aragonite. The split of $v_3$ into 3 peaks observed in coral skeletons is clearly shown in CCHH, whereas $v_3$ splits into 2 peaks in MHC reference standards. The $v_2$ peaks are identically positioned for CCHH and aragonite ($853\,cm^{-1}$), whereas the MHC $v_2$ has a significant shift to higher frequency ($870\,cm^{-1}$). The coral data have only aragonitic or CCHH-like $v_2$ peaks.

Thus, SINS confirmed that coral skeleton surfaces contained abundant CCHH, MHC, $ACCH_2O$, and ACC. Synthetic $ACCH_2O$ and ACC were not significantly different from one another. The exact locations on the coral skeletons from which the spectra in Fig. 4 were extracted are presented in Supplementary Fig. 7, and an example synthetic mineral sample is presented in Supplementary Fig. 8. Notice in Supplementary Fig. 8 that $ACCH_2O$ is blue! The blue color is most visible after compressing $ACCH_2O$ and observing it on a black background. On a white background it appears yellow. X-ray diffraction (XRD) patterns for the synthetic CCHH and MHC standards are shown in Supplementary Fig. 9, confirming the expected crystal structures or lack thereof in amorphous minerals. All XRD spectra for all synthetic samples are presented in Supplementary Data 3.xlsx.

## Discussion
In all biominerals, the four metastable phases always mapped near the forming surfaces, consistent with their being precursors to biomineral

formation. In previous studies these crystalline precursors were not revealed because the analysis was limited to three components including only metastable amorphous precursors, e.g., $ACCH_2O$, ACC, and stable crystalline calcite in sea urchin spines[25]. After finding evidence for metastable crystalline precursors CCHH and MHC, we re-analyzed all previous data on sea urchin spines and found a consistent improvement in fit reduced $\chi^2$ for 5 components compared to the previous 3 components (Supplementary Table 3). Despite this improvement in fit, however, the number of pixels assigned to CCHH or MHC was quite low, less than 5% of precursor pixels for both phases in sea urchin spines (Supplementary Table 5).

Synthetic CCHH and MHC transform into aragonite, not calcite[33,36]. The observation here that aragonitic coral skeletons and nacre had abundant CCHH (-50%) and that calcitic sea urchin spines had only trace amounts (<5%) or no CCHH and MHC is consistent with the synthetic case: CCHH and MHC are not significant precursors to calcite formation in sea urchin spines.

In addition to all four precursor phases on the forming surfaces of nacre and coral skeletons, in nacre, we found CCHH and ACC in discontinuous 100 nm-thick layers at the interfaces between layers of forming tablets (Supplementary Fig. 10). Previous observations showed that the 5 nm-thick surface of nacre tablets remains amorphous even in mature nacre[37]. Here we observed that ACC and CCHH layers were present only between forming tablets, not between mature tablets (Supplementary Fig. 10), and that they were much thicker.

The data in Fig. 3, Supplementary Figs. 4–6 are representative of the entire set of data, summarized in Supplementary Table 5 and Supplementary Data 2.xlsx. Briefly, CCHH was the most abundant precursor phase in nacre and coral, observed at >50% concentration in 48% and 46% of precursor pixels, respectively. ACC was the second most abundant precursor in nacre and coral, found in 36% and 39%, respectively. After that was MHC, found in 9% and 12% of precursor pixels in nacre and coral. MHC only appeared in areas that also contained CCHH. Finally, $ACCH_2O$ was the least abundant precursor phase, detected in only -5% of all precursor pixels in all biominerals. Notably, in nacre 100% of the areas analyzed contained abundant CCHH, whereas in coral skeletons -80% of the areas showed CCHH. Sea urchin spines, on the other hand, contained a completely different combination of precursor phases. ACC was the most abundant precursor phase in spines by an enormous margin at 90% of all precursor phases detected. After that was $ACCH_2O$ at 5%, then CCHH at 3%, and MHC at 2%.

## Phase transitions in time and space
All phases underwent phase transitions, as observed in repeat acquisitions of the same areas. Supplementary Figs. 4–6 show the same areas as Fig. 3's first and second acquisitions (that is, after time and exposure to the beam), so one can compare the same pixel across the 1st and 2nd panels, e.g., looking at specific pixels to see $ACCH_2O$ transform into ACC or aragonite, and CCHH or MHC transform into aragonite.

These data, then, suggest that both CCHH and MHC are transient precursors to aragonite, and that they are intermediate nanoscale carbonate phases between $ACCH_2O$ and the mature stable phase.

The spatial distribution of phases transforming into one another from one cluster of pixels to the next near the forming surface of biominerals may also suggest a sequence of phase transitions. In Supplementary Fig. 11 an example is provided showing gradual transformation of $ACCH_2O \rightarrow CCHH \rightarrow$ aragonite. In Supplementary Fig. 12, the same area as Figs. 2, 3, Supplementary Figs. 4, 11 is displayed as RGB and CMY maps, which enable the display of mixed phases, further highlighting this gradual transformation.

Both lines of evidence, temporal (Supplementary Figs. 4–6) and spatial (Figs. 2, 3, Supplementary Figs. 11, 12), suggest that CCHH and MHC occur as crystalline transient precursors to aragonitic but not to calcitic biominerals. This suggestion makes the process of

biomineralization more complex than previously appreciated, with four possible metastable phases, two amorphous and two crystalline. The only crystalline precursor phase previously observed was vaterite, found as a precursor to calcite formation in foraminifera biomineralization[38]. In all three biominerals analyzed here, vaterite was never observed to significantly improve fitting in MMs in areas of any forming biominerals (Fig. 1, Supplementary Table 3, Supplementary Data 2.xlsx). X-ray absorption spectra of ikaite remain elusive because the mineral is not stable at room temperature, thus, we could not test if ikaite is a precursor phase, but this hydrated crystalline phase has only been observed to form in near-freezing environments[39], making its presence unlikely in any of the biominerals studied here. We therefore suggest that CCHH and MHC are the most likely crystalline intermediates in coral skeleton and nacre biomineralization.

## A new energy landscape for $CaCO_3$ biomineralization
More phase transitions were introduced by the observation of CCHH and MHC precursor phases, thus, the energy landscape of the biomineralization process became more complex, including several possible pathways. Using previously measured enthalpies and entropies in synthetic phases[40–43] and sequences of phase transitions[33], it was possible to construct a hypothetical energy landscape with all the phases observed in biominerals. This energy landscape included four possible pathways to mature biomineral formation for aragonite biominerals: $ACCH_2O \rightarrow$ aragonite, $ACCH_2O \rightarrow$ ACC $\rightarrow$ aragonite, $ACCH_2O \rightarrow$ CCHH $\rightarrow$ aragonite, or $ACCH_2O \rightarrow$ MHC $\rightarrow$ aragonite. Conversely, the landscape for calcite biomineralization was simpler, with only two pathways to mature biomineral: $ACCH_2O \rightarrow$ calcite, or $ACCH_2O \rightarrow$ ACC $\rightarrow$ calcite. These energy landscapes are presented schematically in Fig. 5.

The kinetic barriers that govern two of the phase transitions, dehydration of $ACCH_2O$ and crystallization, were previously measured[43]. Water plays a significant role in controlling the height of these barriers. The height of the other barriers is unknown, thus, for simplicity, they were plotted with similar heights in Fig. 5, save the barrier for the direct transition from $ACCH_2O$ to mature biomineral, which must be higher otherwise the observed diversity of phases would not be detectable.

During biomineralization, the more complex the energy landscape, the greater the number of phase transitions necessary to reach the final stable polymorph, the more biological control the organism has over the crystallization process. This control, and indeed the existence of the different pathways in Fig. 5, may be mediated by organic molecules, especially proteins. In sea urchin biominerals, for example, the SM50 protein slows down the dehydration of $ACCH_2O$ to ACC[21]. Catalysis is also possible, rather than inhibition of phase transitions[44]. A complicated system of phase transitions may provide a benefit to organisms: by gaining control over the crystallization process, organisms more effectively direct it towards the crystal polymorph, morphology[45], size, and crystal arrangements[46] that provide them with better function and therefore greater evolutionary advantage.

In synthetically prepared CCHH in aqueous conditions, water is gained or lost: during the transition from $ACCH_2O$ to CCHH, ½ $H_2O$ per formula unit is lost, during CCHH to MHC, ½ $H_2O$ is gained[33]. The latter hydration, therefore, must be thermodynamically downhill. Within our experimental conditions, we rarely observe biogenic CCHH transforming into MHC (cyan to magenta transition, 2% of precursor pixels, Supplementary Table 5B), therefore that pathway is excluded from Fig. 5. Similarly, in Fig. 5 we did not include the transition from ACC to CCHH because we rarely observe it (green to cyan, 2%). Mechanistically, any transition from ACC to CCHH requires hydration in a solid, a process that is kinetically inhibited, and may not happen on the time scale of biomineralization (1–2 days). By contrast, all other thermodynamically downhill transitions indicated by arches in Fig. 5 occurred in most precursor pixels analyzed ($\geq$87%).

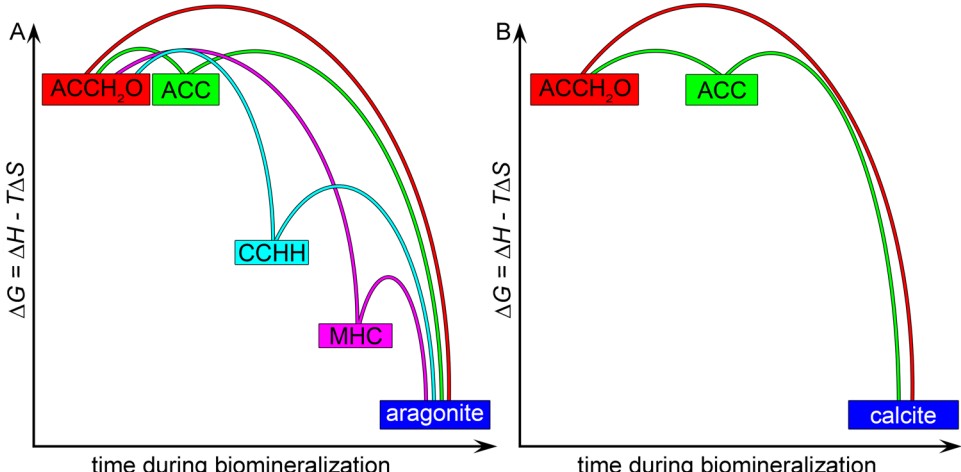

**Fig. 5 | Possible energy landscape leading to the formation of aragonite or calcite biominerals.** The landscape includes all phases thus far found during biomineral formation: metastable $ACCH_2O$, ACC, CCHH, MHC, and stable aragonite or calcite. **A** There are four pathways to crystallization from $ACCH_2O$ to a mature aragonite biomineral: direct or through ACC only, the other two pathways include one at a time CCHH or MHC. **B** In contrast, there are only two pathways from $ACCH_2O$ to a mature calcite biomineral, either directly or through ACC. $ACCH_2O$ and ACC are isoenergetic[43] and most metastable, all other phases are downhill in the order observed in repeat acquisitions in biominerals here (Supplementary Table 5B) and previously in synthetic systems[40–43]. The color coding is the same as for all spectra and MMs in this work.

A multi-step crystallization pathway will impact the elemental and isotopic compositions of biominerals, and thus the results of paleoclimate reconstructions based on these geochemical proxies[7,47,48]. Compositional differences between synthetic and biogenic minerals as well as amongst different biominerals formed in similar conditions[49,50] may be in part due to the presence of multiple precursor phases during biomineral formation.

**Solid-state transformation or dissolution and reprecipitation?**
If all phase transitions (solidification to $ACCH_2O$, dehydration, crystallization to CCHH, MHC, aragonite, or calcite) occur as solid-state transformations[51], the composition of the final mineral must be governed by the first precursor phase. If instead the phase transitions occur by dissolution and reprecipitation[26,52], there are not one but multiple opportunities for exchange of ions and isotopes with the surrounding solution[48,53]. Exchange occurring at each phase transition would then depend on the physical and chemical characteristic of the calcifying fluid, and those of each solid phase, e.g., the elemental and isotopic affinities of each phase, the concentrations and residence times of ions and isotopes in the calcifying fluid, the volume of the fluid, or the $CaCO_3$ precipitation rate. Thus, even if there is exchange, the effect of the surrounding fluid may or may not be significant. These unknowns warrant further investigations in natural forming biominerals in living animals. Furthermore, experiments in synthetic settings could clarify the effect of precursors and dehydration on trace element and isotope compositions in the final crystalline mineral[54].

Since all precursor phases, amorphous and metastable crystalline, were always observed at the forming surface of all biominerals, phase transitions certainly have access to water, thus, it is entirely possible that dissolution and reprecipitation occur. For example, in the coral extracellular calcifying fluid the supersaturation was measured with micro-electrodes to be $\Omega_{arag} \cong 12$[24,55]. Such supersaturation allows for the dissolution of precursor carbonates but not aragonite dissolution. However, if this was the only mechanism, in the solid skeleton the metastable precursor phases observed here, amorphous and crystalline, would never be observed. Their observation implies unequivocally that at least some precursor particles never dissolve and reprecipitate, but undergo solid-state transformation.

Sun et al. argued that in corals both mechanisms must occur based on the porosity of the surface layer observed in PEEM experiments and the space filling nature of all mature coral skeletons[24]. In nacre, instead, Otter et al.[26] showed that the dissolution and reprecipitation mechanism dominates. Our data certainly do not contradict theirs. Our observation of metastable precursors at the forming surface of nacre may be interpreted in two different ways. In one scenario, some precursor particles dissolve more slowly than others, and those slow-precursors are observed at the surface. They eventually dissolve and reprecipitate, thus, they never undergo a solid-state transformation. In another scenario, the precursor particles never dissolve but undergo a solid-state transformation. We do know that it is possible to crystallize synthetic $ACCH_2O$ into calcite via a solid-state transformation, as detected by NMR spectroscopy[56].

One could speculate that perhaps both mechanisms could occur in various biominerals, but to different extents, or at different locations or formation stages. Further studies, designed to tease these two mechanisms apart, will shed light. The present study was not designed to address these mechanisms thus, it cannot settle the debate.

**Potential artifacts**
Here we observed metastable crystalline CCHH and MHC on the aragonite or calcite skeleton surfaces of diverse biominerals. This observation is consistent with previous evidence for amorphous minerals in intracellular vesicles as the starting venue for biomineralization across phyla[16,47,57–60]. The amorphous phases in this work, in fact, appeared at the outermost surface of skeletons. What the present data did not show is a time sequence for the biomineralization process. A potential artifact we could not exclude is the possibility that some crystalline phases observed on the skeleton surface crystallized after the death of the animal, either during sample preparation or during data acquisition. In this scenario, it is possible that in the living animal, all phases are amorphous in intracellular vesicles and on the skeleton surface, and crystallization only occurred after the animal's death and before our analysis (~1 day after death). In a few cases where a direct comparison was possible, comparing the sample preparations and results from the x-ray spectromicroscopy used here yields similar results. For example, entire sea urchin spicules in entire embryos[61] or extracted, polished, and coated spicules[21] gave consistent results. Intracellular vesicles in an entire spicule-forming cell[57,60] and those surrounding an embedded and polished sea urchin spine[58] gave intracellular vesicles consistent in shape, size, and composition. However, it is impossible to

rule out that some phase transitions could have occurred during sample preparation for MMs or SINS-FTIR. Even if more or faster crystallization occurred due to sample preparations, it is safe to assume that the amorphous phases were originally in intracellular vesicles, were then deposited by the living animal on the surface of the forming biominerals, and they crystallized into progressively more stable polymorphs. Potential artifacts may have pushed the system thermodynamically downhill, as observed before during radiation damage studies[21] with significantly greater dose than used here. Thermodynamically downhill transitions are in the top-to-bottom direction in Fig. 5, such as faster dehydration or crystallization.

Synthetically, transformation of $ACCH_2O$ into CCHH, then MHC, and finally aragonite has been ascribed to dissolution and re-precipitation in the presence of water[33]. Biogenically, these may be mostly solid-state transformations, and occur in the absence of bulk water. In repeat acquisitions, we observe these transitions in the solid state, in the absence of water, and in ultra-high vacuum (UHV) (Supplementary Figs. 4–6). This may happen exclusively in UHV and therefore be an artifact, or it may be what happens in the forming skeleton in the living animal. The question remains open. But again, we push the system towards crystallization of the most stable polymorph in 70% of the 190,000 precursor pixels analyzed that undergo transitions, and >99% of the 200 million calcium containing pixels analyzed do not undergo any transition.

Another potential artifact is differential dissolution of aragonite vs. calcite biominerals due to varying organic molecules. These are certainly different in each organism, and they affect solubility and therefore potential dissolution artifacts differently in each biomineral.

Even if one or more of the above artifacts partly affected the results presented here, they unlikely generated phases that were not also present in the living animal.

## A bright crystalline future

One important experiment in the future will be the observation of aragonite and calcite biominerals formed simultaneously by the same organism, for example mollusk shell's nacre aragonite and prismatic calcite at the forming lip of either bivalve or gastropod shells. If these experiments demonstrate that CCHH and MHC precursors exist in aragonite but not calcite forming biominerals, it would provide even stronger and less artifact-prone evidence than presented here. Other interesting experiments could explore the effect of proteins or additives (e.g. Mg) on selection of the different pathways to aragonite or calcite.

One analytical key advantage of crystalline vs. amorphous precursors is that future studies can explore their presence with electron diffraction methods, e.g., Electron Back-Scatter Diffraction (EBSD). Furthermore, because CCHH has ½ $H_2O$ molecule per formula unit, it is distinguishable from all other polymorphs by mass-spectrometry, e.g., nano-Secondary Ion Mass Spectrometry (nano-SIMS). Both approaches could explore polished cross-sections of freshly deposited biominerals as done here with x-ray spectromicroscopy, and both have the resolution and surface-sensitivity necessary to detect these nanoscale transient precursors. Photo-induced Force Microscopy (PiFM), with 5 nm lateral resolution and about 5 nm depth sensitivity in sideband mode, is also sensitive to carbonate FTIR spectroscopy[62]. Importantly, EBSD, nanoSIMS, and PiFM are commercial instruments vastly more widely available than the synchrotron PEEM and SINS approaches used here. Thus, the unexpected result presented here can generate the wide-spread studies necessary to crack open the mysteries of biomineralization, from cells to skeletons.

## Methods
### Coral skeletons

Colonies of tropical corals, *Stylophora pistillata* were grown in the long-term culture facilities at the Centre Scientifique de Monaco (CSM)

in aquaria supplied with Mediterranean seawater, at constant temperature of 25 °C, salinity of 38 ppt, light exposure of 200 μmol photons $m^{-2}$ $s^{-1}$ (=10,800 Lux), supplied twice weekly with *Artemia salina* nauplii and marine food (Ocean Nutrition, Newark, CA, USA).

For each experiment, 10 *Stylophora pistillata* coral nubbins (2 cm in length) were prepared. Each experiment was repeated 5 times, one week apart, also at CSM, and shipped to the US for analysis. The day of each experiment 10 nubbins were removed from the aquarium and were:

- immersed into seawater enriched with 5 w% $MgCl_2$ for 30 min.
- fixed for 30 min in 2% paraformaldehyde in Buffer 1
- rinsed 2 × 10 min in Buffer 1
- immersed into increasing concentrations of ethanol (EtOH, 50%, 60%, 70%) in Buffer 2, 3 × 10 min.
- immersed into increasing concentrations of ethanol (80%, 90%) in Buffer 3, 2 × 10 min.
- immersed into ethanol 100% 2 × 10 min.
- once dehydration was complete, the samples were cut into small pieces, -1 cm long, using bone cutting forceps
- immersed immediately in a bath of 75 v% EpoFix + 25 v% EtOH for 30 min. (EpoFix from Electron Microscopy Sciences, Hatfield, PA, USA).
- immersed in two EpoFix baths before pressurizing them to 2 atm in a third EpoFix bath.
- cured overnight at 2 atm pressure, which increased the temperature of sample from 21 to 23 °C.
- shipped by courier to either Madison, WI or Berkeley, CA.

The recipes for preparing Buffers 1, 2, 3 are as follows.
- Prepare 22 g/L $Na_2CO_3$ in Milli-Q water; filter through 0.2 μm filter. Keep 1000 mL as is. Dilute the rest to obtain 200 mL 1 g/L $Na_2CO_3$ solution, and 200 mL 0.5 g/L $Na_2CO_3$ solution.
- Buffer 1: 0.05 M Na Cacodylate buffer in 22 g/L $Na_2CO_3$; add 10.7 g Na Cacodylate powder to 1000 mL 22 g/L $Na_2CO_3$ solution.
- Buffer 2: 0.002 M Na Cacodylate buffer in 1 g/L $Na_2CO_3$; add 0.086 g Na Cacodylate powder to 200 mL 1 g/L $Na_2CO_3$ solution.
- Buffer 3: 0.002 M Na Cacodylate buffer in 0.5 g/L $Na_2CO_3$; add 0.086 g Na Cacodylate powder to 200 mL 0.5 g/L $Na_2CO_3$ solution.
- Fixative: 2% paraformaldehyde in Na Cacodylate buffer; add one 10 mL ampule of 16% Formaldehyde to 70 mL of Buffer 1.

It is important to prevent dissolution or (re)crystallization of any amorphous or metastable crystalline precursor phases in coral skeletons during sample preparation. Thus, all sawing, grinding, and polishing solutions contained a concentration of $-CO_3$ much greater than the dissolution concentration of ACC. These are, respectively, 22 g/L and 6 mg/L. No Ca was ever added to any of these solutions, thus, all the Ca analyzed in biominerals in this work was originally present in the biominerals.

All corals were first immersed in seawater enriched with 5 w% $MgCl_2$ to relax the tissues, and prevent the retraction reaction that polyps exhibit when touched.

The cacodylate buffer was used during fixation in paraformaldehyde because it helped better preserve the tissue, which remained attached to the biomineral growth front, and therefore protected the delicate metastable precursor phases at the biomineral surface.

Corals and all other samples described here were dehydrated in ethanol because ethanol binds to ACC and all other carbonates[63], therefore protecting metastable phases from transforming[21,23,24,64]. A protective layer of ethanol before embedding better preserves fresh samples. Right before beamtime, the polished surface is also soaked with ethanol, so the metastable precursors are stabilized by a surface monolayer of ethanol.

All other coral samples were described in previous publications[23,24,65]. None of the data presented in any

figures here were previously published. The older data were re-processed to explore the prevalence of CCHH in other species. All previous corals were fixed, dehydrated, and embedded identically to the new ones above. These were *Stylophora pistillata*[33], *Acropora* sp., *Blastomussa merleti*, *Micromussa lordhowensis*, *Montipora turgescens*, *Turbinaria peltata*, and again *Stylophora pistillata*[24], and *Acropora* sp., *Stylophora pistillata*, and *Turbinaria peltata*[65].

## Nacre
Ten California red abalone *Haliotis rufescens* shells were purchased alive from the Monterey Abalone Company (Monterey, CA, USA), kept alive in a seawater aquarium at 15 °C, then sacrificed. The animal was removed, and the shells were then cut into 1 cm × 1 cm pieces, near the lip, where new nacre deposition always occurs. The shell pieces were fixed, dehydrated, and embedded as described above and in a previous paper[22].

## Sea urchin spines
Thirty regenerating sea urchin spines from the 10 California purple sea urchins *Strongylocentrotus purpuratus* three spines from one *Paracentrotus lividus* were obtained by cutting 3 spines per urchin with scissors, then letting them regenerate for 7–10 days in an aquarium at 15 °C, feeding them fresh kelp regularly. After regeneration, entire spines were fixed, dehydrated, the most mature part of each spine was cut so that the regenerating parts were retained and 4 spines fit into 1-inch round molds, and embedded as described above and in a previous paper. Figure 2 in Albéric et al.[25] shows the regenerating and mature parts of 4 spines after embedding and polishing.

## Synthetic carbonates
Inorganic calcite, aragonite, ACCH$_2$O, ACC, and MHC samples used in SINS-FTIR analysis were synthesized at the Lawrence Berkeley National Laboratory (LBNL) by one of us (LSD), whereas CCHH was synthesized by another one of us (ZZ) at Wuhan University. For the samples synthesized at LBNL, Ca-MgCl$_2$ and Na$_2$CO$_3$ solutions of different volumes, concentrations, and temperatures were mixed to obtain the desired polymorphs[33,66] (Supplementary Table 6). Following CaCO$_3$ precipitation of the target phase, the solutions were filtered under vacuum using cellulose acetate filters of 0.45 μm pore size. The stable phases (calcite and aragonite) were rinsed with Milli-Q water to remove any NaCl whereas the unstable phases (ACCH$_2$O and MHC) were rinsed with absolute ethanol to remove surficial water and prevent dissolution and reprecipitation into other phases. X-ray diffraction (XRD) analysis confirmed the phase and purity of each sample (see XRD method section). The dehydrated form of ACC was obtained by heating ACCH$_2$O in an oven to a temperature of 200 °C for more than 2 h. The latter procedure was guided by anterior Thermo-Gravimetric Analysis (TGA) and Differential Scanning Calorimetry (DSC) data obtained from the same ACCH$_2$O sample (Supplementary Table 6). All samples were stored in a vacuum desiccator between the various analyses. For powders, crystalline and amorphous, for SINS-FTIR analysis were pressed all powders into carbon tape (Ted Pella, Redding, CA) using a KBr press, then analyzed the surface with SINS-FTIR. These are shown in Supplementary Fig. 8.

Inorganic calcite, aragonite, and vaterite used in PEEM analysis for the component spectra were synthesized at the University of Wisconsin-Madison by another one of us (CAS). Aragonite crystals were produced using the ammonium carbonate diffusion method described in previous work[67], in which a 50 mL solution of 10 mM CaCl$_2$ and 50 mM MgCl$_2$ was divided into three beakers covered in aluminum foil with six holes poked in the top, then placed in a sealed desiccator alongside a partially covered beaker of solid ammonium carbonate ((NH$_4$)$_2$CO$_3$) and left to sit sealed at room temperature for 24 h. Crystals formed at the bottom of the beakers were collected by scraping

with a plastic spatula, rinsed with ethanol, and dried. Calcite and vaterite crystals were produced with the same procedure, only the solution contained 5 mM CaCl$_2$ and no MgCl$_2$. The crystals at the bottom of the beaker were a mix of calcite rhombohedra and approximately round vaterite particles which were easily distinguished morphologically in PEEM for the extraction of spectra from the two different phases. All sample compositions were confirmed with powder XRD. All synthetic crystals for PEEM analysis were embedded in EpoFix as (EMS, Hatfield, PA), then polished and coated as described below.

## Sample preparation
All biomineral samples embedded in 1-inch rounds of EpoFix (EMS, Hatfield, PA, USA) were cut to expose the area of interest of the forming biomineral immediately before beamtime experiments. They were cut using a low-speed diamond saw (TechCut 4, Allied Inc., Rancho Dominguez, CA, USA) and 22 g/L Na$_2$CO$_3$ in DI water as a coolant. They were then polished using an automated high precision polisher (Automet 250 Pro Grinder Polisher, Buehler, Lake Bluff, IL, USA) using 320, 600, 800, 1200, and 4000 grit SiC paper for 1 min each with a force of 5 N, and 40 rpm for the head and 140 rpm for the platen, rotating in the same direction. Then we switched to MicroFloc (Buehler) velvet on the platen, and 300 nm Al$_2$O$_3$ nanoparticles (MicroPolish, Buehler), then 50 nm Al$_2$O$_3$ nanoparticles (MasterPrep, Buehler) for 2 min each with a force of 10 N, and 40 rpm for the head and 140 rpm for the platen, rotating in opposite directions. During grinding and polishing we only used 22 g/L Na$_2$CO$_3$ in DI water as a coolant and lubricant, never DI water, to avoid dissolving transient precursor phases. The 300 nm Al$_2$O$_3$ nanoparticles were suspended in suspension in 22 g/L Na$_2$CO$_3$ in DI water, whereas the MasterPrep, which comes as a liquid, was dialyzed against 22 g/L Na$_2$CO$_3$ in DI water 3 times in 24 h.

We then sonicated the samples upside down in EtOH, to remove any polishing nanoparticles, and cleaned the sample upside down on TexWipe® (TX309 Dry Cotton Cleanroom Wipers, TexWipe, Kernersville, NC, USA) placed on Carrara marble, and soaked in EtOH, never re-using the same portion of TexWipe.

All samples for PEEM analysis were coated with 1 nm Pt in the area of interest and 40 nm Pt all around it, to make the electrical contact necessary for the sample to be at high voltage for analysis. This was achieved using a high resolution Cressington 208HR (Cressington, UK, Ted Pella, USA) sputter coater with a rotary and planetary tilt stage. The area of interest was covered by a diced Si wafer 4 mm × 4 mm, the surrounding area was coated with 40 nm Pt with the sample static and not tilted. The chamber was vented, the Si wafer removed, then the sample was coated with 1 nm Pt while tilted and spinning at the max speed possible to guarantee homogeneity of the coated thin 1 nm layer of Pt[35,68,69].

Importantly, the two coral skeletons analyzed with SINS-FTIR were prepared differently: first sample (Sp67pH8, Supplementary Fig. 7C, D) was cut, ground, and polished using 22 g/L Na$_2$CO$_3$ solution as a coolant, the second (Sp65pH8, Supplementary Fig. 7A, B) using 25 g/L CaCl$_2$. The resulting spectra were identical, thus, none of the $v_2$ or $v_3$ peaks relative to C-O bonds observed in coral samples originated from the cooling Na$_2$CO$_3$ solution.

Similarly, none of the PEEM data are affected by sample preparation, because only Na$_2$CO$_3$ solution was used, which does not contain any Ca, and all PEEM data are acquired at the Ca L-edge.

## PEEM analysis
We used the PhotoEmission Electron Microscope (PEEM) at the Advanced Light Source, Lawrence Berkeley National Laboratory, named PEEM-3, on beamline 11.0.1.1.

All data were acquired at the Ca L-edge, which is where the calcium carbonates are most distinct from one another, and thus is best

for mesoscale carbonate phase analysis of fresh, forming biominerals. For each Ca L-edge acquisition we acquired a stack of 121 PEEM images, each with more than $10^6$ pixels ($1030 \times 1054 = 1,085,620$ pixels), while scanning the photon energy between 340 eV and 360 eV, with 0.1 eV resolution between 345 eV and 355 eV where Ca L-edge peaks exist, and 0.5 eV steps elsewhere. The polarization was set to be circular to minimize the effect of crystal orientations and dichroism. Each pixel of the Ca acquisition contains the full Ca L-edge spectrum of the unknown phases present in that pixel, and can be acquired identically in mature biominerals, on the fresh, forming surface to analyze transient precursor phases, the embedding EpoFix surrounding the biomineral, and also the tissue depositing the biomineral at the time it was fixed. The lateral resolution is at best 20 nm on PEEM-3, although frequently we acquired Ca acquisitions with 60 nm pixels to maximize signal, minimize exposure time and radiation damage. The detection thickness at the Ca L-edge is only 3 nm[35] in Figs. 1, 2, 3, Supplementary Figs. 4–6, 10–12.

Regions analyzed were always virgin, meaning never exposed to the x-ray beam before analysis, to prevent radiation damage. Repeat acquisitions show slight variation in a few pixels, most frequently thermodynamically downhill, with amorphous phases becoming more crystalline, and CCHH becoming aragonite. These results are summarized in Supplementary Table 5B.

Each Ca acquisition was acquired in duplicate in each area, and we acquired a total of 54 areas of coral skeletons from 3 different *Stylophora pistillata* skeletons and from 4 other corals, 16 areas of nacre from 2 *Haliotis rufescens* shells, and 16 areas of sea urchin spines from 4 different *Strongylocentrotus purpuratus* sea urchins and 1 *Paracentrotus lividus*. One or two areas per species, acquired in duplicate, would be sufficient to draw conclusions, thus, the sample size analyzed here, is greater than necessary for statistical significance and it also included biological variability.

## Obtaining the Cni16 carbonate phase component spectra

The Cni16 spectra, listed in Supplementary Table 1, displayed in Figs. 1, 2, Supplementary Fig. 1, and used for all Myriad Maps (MMs) mesoscale mineral-phases of carbonates presented here, were obtained as follows.

1. All Ca L-edge spectra were acquired on the same PEEM-3 microscope on beamline 11.0.1.1 at ALS, using the medium energy grating (MEG) and the elliptically polarizing undulator in the 1st harmonic, with ALS operating at 1.9 GeV, with circular polarization (pol 1) to minimize crystal orientation effects.
2. All spectra were obtained from synthetic carbonates.
3. All single-pixel spectra were aligned in energy and amplitude, so that peak 1 was at 352.6 eV and had intensity 10.
4. All single-pixel spectra averaged, again, to minimize the effect of statistical and non-statistical noise.
5. Each average spectrum was peak fitted, to further eliminate noise. We used precisely the same 3rd order polynomial background, and the same 2 arctangent $L_2$ and $L_3$ absorption thresholds for all spectra to eliminate background effects. All Cni16 spectra are presented in Figs. 1, 2, Supplementary Fig. 1, the list of component spectra and their color coding is presented in Supplementary Table 1, peak fitting coefficients are presented in Supplementary Table 4.

The Cni16 spectra are provided in file Supplementary Data 1.xlsx. These spectra are consistent with previously published spectra from synthetic, geologic, and biogenic samples[23–25,33,64,65]. With three main differences:

1. We forced all spectra in Cni16 to share the same background and arctangents, thus, preventing background-related mapping artifacts, which have no physical meaning. All spectra were acquired under identical conditions and on the same beamline. The beamline background intensity is completely flat and featureless at the Ca L-edge energies; thus, this choice is justified.
2. We normalized all spectra in Cni16 to have the same intensity 10 on peak 1, before peak fitting. This eliminates artifacts due to lower density and therefore Ca concentration at the porous surface of biominerals.
3. The spectra in Cni16 originated from synthetic minerals, whereas component phase spectra in previous work were biogenic in origin[22–24,65]. CCHH has only previously been observed synthetically, never biogenically, thus, the only option to detect CCHH was to use synthetic spectra. For consistency, therefore, we used all synthetic spectra for all phases.

Comparing biogenic (Cni14) and synthetic (Cni16) spectra, we observed that the fit to aragonite minerals and biominerals improved significantly using synthetic Cni16 spectra, whereas for calcite minerals and biominerals the fit did not vary significantly (synthetic), or got worse (in mature sea urchin spines). The results of this comparison are presented in Supplementary Table 2. For consistency of analysis across all samples, we used the synthetic calcite spectrum.

## Peak fitting

Using the GG Macros[70], all spectra were aligned, including energy shift, in the energy range that includes the peaks, from 345 to 355 eV. The peak fitting results are presented in Supplementary Fig. 2 for two representative minerals, calcite and CCHH.

1. A linear fit to the pre-edge was subtracted from each spectrum.
2. Arctangents for calcite were found, placed 0.25 eV below peak 3 and below peak 1, fixed in position, width, and amplitude, and re-used identically for all spectra of all minerals. There is no reliable way to physically measure or precisely calculate with modeling, e.g., FEFF[71], where the arctangents are, therefore, this choice is somewhat arbitrary, but it is made consistently across all spectra of all minerals.
3. A polynomial background (p0-p3) for calcite, then re-used identically for all spectra and kept fixed during peak fitting. This was key to obtaining a consistent pre- and post-edge background for all spectra.
4. All spectra were rigidly shifted until peak 1 was at exactly 352.6 eV, then the maxima of peaks 1 and 3 were adjusted manually to be perfectly symmetric around the maximum if the very tip (1 or 2 data points) of a peak happened to be asymmetrical due to sampling with 0.1 eV step during data acquisition and the physical peak not falling precisely on one of the energies spaced 0.1 eV apart. For example, if a peak is at 352.63 eV but we only sample 352.6 and 352.7 eV, the latter 2 datapoints would not be at the same intensity. In this case we manually adjusted the intensity, keeping the highest intensity point as the center of the peak, and manually moving the intensity of the 2 datapoints before and after it up or down to make the peak as symmetric as possible around the center. This minimal adjustment only affected 2 data points out of the 121 present in each spectrum, and it never shifted the energy or intensity of a peak.
5. The intensity of all spectra was adjusted to 0 and 10, pre-edge and peak 1, with left cursor in GG Macros at 345 eV and right cursor at 352.6 eV. The left and right cursors define these two positions: pre-edge at 345 eV, 352.6 eV on top of peak 1.
6. Each spectrum was peak fitted starting from the fit coefficients of the previous spectrum, freeing 1 fit parameter at a time, thus, peak-fitting slowly to avoid exploding, which happens when too many fit parameters are freed at once when the fit is far from the experimental spectrum. Slowly fitting one parameter at a time until the experimental and fit spectra are similar, makes it possible to free many parameters at once. Supplementary Fig. 3 shows that

the order in which peaks are feed does not affect the result of peak fitting.

7. Peaks that did not fit well were isolated and fit only within the energy range of that peak, excluding overlap with neighboring peaks. This was always necessary for peak 2 because of a significant intensity dip between peaks 2 and 3 in all spectra. Isolating peak 2 and not allowing the software to fit it to the dip made it possible to fit the lineshape of peak 2 itself perfectly (zero residue). Some peaks required fitting by trial and error while holding parameters, not letting them freely vary, to force the software to recognize important sections of the lineshape. This was necessary in energy ranges with heavily overlapping peaks such as the aragonite peaks 2 and 2' or peaks 4, 4', and 4", as shown in Supplementary Table 4, to avoid the fitting peaks in non-physical ways. For instance, during fitting with completely free peaks 2 and 2', peak 2 was fit with a reasonable amplitude and width, and matched the data reasonably well, whereas peak 2' had enormous width and/or very small amplitude, which helps the fit outside the region of peak 2' but not on peak 2', where the fit and the data did not match at all.

8. The overlapping peaks in the peak 4 region of CCHH also required holding peak parameters during fitting, repeating step 7 with just one peak at a time for peaks 4, 4', 4", and 4"'.

9. Finally, the relatively broad peaks 2 and 4 for CCHH, MHC, and aragonite were slightly exaggerated manually in intensity, to ensure that fits of unknown spectra to these component spectra took into account the precise positions of these peaks. This is clearly shown in the CCHH peaks 4', 4", and 4"' in Supplementary Fig. 3.

10. This produced the Cni16 component spectra listed in Supplementary Table 1, peak fitted in Supplementary Table 4, and presented in Figs. 1, 2, Supplementary Fig. 1.

## Decisions made during data processing

Myriad Mapping (MM) mesoscale carbonate-phases from spectromicroscopy data requires multiple choices, which are much better made by people than automatically or by any machine learning (ML) or artificial intelligence (AI) system. We estimate that automating these decisions, thus, rigorously exploring the entire parameter space for each decision, will take 2–4 years of continuous operation for a 10 TFLOPS (tera floating point operations) computer (e.g. an Apple M2 Max CPU, or an AMD Ryzen 5, both with ~3.5 GHz, and 10–100 GB of RAM).

Furthermore, parallel processing avoids preconceived biases and mistakes, and their inadvertent propagation by ML systems. The parameter space is too large for ML at this stage; thus, decision-making and data quality judgement is much better done by clever people than by machines. Convergent solutions for each data set were obtained for all biominerals and all 86 areas analyzed in duplicate and triplicate here.

MM identifies and displays pixels in an acquisition that contains component spectra, that is, whose composition matches that of a previously known mineral phase that has a characteristic spectrum. In this work, all acquisitions were acquired across the Ca L-edge using PEEM and were therefore 1 Ca L-edge x-ray absorption near edge structure (XANES) spectrum of unknown composition for each pixel in the acquisition. MMs were produced using the GG Macros, developed for this purpose and distributed free of charge to any interested users on our website.

Each unknown single-pixel spectrum was fit, using non-linear least square analysis, to identify the best linear combination of component spectra that best fits the single pixel spectrum. To obtain the resulting MMs we fitted all million spectra with either 3, 4, or 5 component spectra. The component spectra used here in all analyses were Cni16, shown in Figs. 1, 2, Supplementary Fig. 1. The goodness of the fit was measured in each pixel by the reduced $\chi^2$ of the fit. Reduced $\chi^2$ was calculated from the $\chi^2$ values generated by the GGMacros using Eq. 1,

$$\chi^2_{reduced} = \chi^2_{GGMacros} * \frac{F}{F - N} \tag{1}$$

where $F$ is the number of degrees of freedom and $N$ is the number of components used to generate the fit[72]. The GGMacros calculate $\chi^2$ using the number of data points in the spectrum analyzed thus the degrees of freedom are $F = 110$ points, between 340 and 355, and using $N = 5$ components, thus, to convert $\chi^2$ to the reduced $\chi^2$ multiplication by $F$, then division by $F$ minus $N$ was done for all comparisons here. One of the 4-component analyses used vaterite spectra, which gave only random distributions of vaterite single pixels, equally likely in biominerals or EpoFix areas. Mathematically, the more components a fit has the better the fit, in terms of improved $\chi^2$, and this is a general statement, whether the corresponding mineral phases exist of not. Thus, 4-component carbonate phase analysis is better than 3-. For this reason, we compared all possibilities for 4-carbonate phase analysis:

ACCH₂O, ACC, CCHH, Aragonite
ACCH₂O, ACC, MHC, Aragonite
ACCH₂O, ACC, Vaterite, Aragonite
The final component was of course calcite, not aragonite, for sea urchin spines.

The results of all reduced $\chi^2$ comparisons are presented in Supplementary Table 3. Notice that the addition of vaterite does not significantly improve the fit in pixels that are fit as vaterite ($p = 0.63$) by 6cmp analysis, and that the addition of vaterite in 6cmp analysis does not significantly improve the fit compared to 5cmp excluding vaterite for any pixels assigned to any phase. This difference makes it clear that pixels assigned to vaterite are described just as well if not better by the amorphous precursors, CCHH or MHC, while vaterite does not describe CCHH or MHC pixels well ($\chi^2_{4cmpCCHH} = 0.0069$ vs $\chi^2_{4cmpVaterite} = 0.0084$ in CCHH pixels, $\chi^2_{4cmpMHC} = 0.0083$ vs $\chi^2_{4cmpVaterite} = 0.0105$ in MHC pixels). See these and all other comparisons in Supplementary Table 3. The best analysis is indubitably the one with five components, including both CCHH and MHC ($\chi^2_{5cmpCCHH\&MHC} = 0.0069$ and $0.0083$ in CCHH and MHC pixels, $p = 3.0 \times 10^{-207}$ and $8.0 \times 10^{-86}$ compared with both three components respectively).

The result of MM analysis with 3, 4, or 5 components is one proportion map (pMap) for each component used. In a pMap, each pixel has a value (a number from 0 to 1) corresponding to the proportion of that component that produced the best fit for that pixel unknown spectrum.

There are 12 different decisions to make, to obtain a robust MM, and this is why this complex analysis is best done by many people in parallel, rather than by a machine. For each MM analysis in Igor, using the GG Macros[70] a user decides:

1. when a stack of images is well aligned (e.g., within ½ pixel for lateral motion), and what method to use in aligning a misaligned stack. Alignment is judged in the PeemVision software package, which allows the user to view the stack of images as a movie and watch for drift of features. The user then identifies a distinct feature that stays constant across all images, and uses one of various alignment algorithms (align, align vertical, align horizontal, align 4x oversample, etc.) to align that feature across all images.

2. which component spectra to select for its analysis (3, 4, 5 components, ending with either calcite or aragonite depending on the biomineral)

3. within which energy range the unknown spectrum in each pixel should be fit by a linear combination of the selected component spectra. After many trials, we converged on the energy range 340–355 eV. This includes all of the pre-edge, and all of the Ca peaks, but it excludes the post-edge, because the intensity of the featureless post-edge is directly proportional to Ca concentration

in each pixel, whereas the lineshape in the energy region joining the highest energy peak 1 and the flat post-edge (354–355 eV) is strongly mineral-dependent and should therefore be included in mineral assignment analysis.

4. within what energy interval each spectrum should be allowed to be rigidly shifted in energy (e.g., $\delta E = 0.1$ eV, 0.2 eV, 0.3 eV).

5. If all the above parameters are satisfactory, they start MM, which is automated in GG Macros and completed in 4–10 h. The precise time strongly depends on the above choices, but it is always withing the 4–10-h range. Once the MM analysis is done, the user decides:

6. at what energy is best to place cursors to obtain a Ca map termed diff map. The cursors define on-peak and off-peak energy positions, then Igor subtracts the off-peak images from on-peak images, to obtain difference maps e.g., 352.6 eV for the most intense Ca peak 1 and 343 eV for pre-edge, in Fig. 2, Supplementary Fig. 1.

7. how many images to average on-peak and off-peak to obtain the most informative diff map (e.g., 5 on-peak and 7 off-peak), then click a subtract button to calculate and display the diff map.

8. where to place a threshold on a diff map (e.g., 20), based on how noisy Ca spectra are. If the Ca spectrum in a pixel is too noisy that pixel should be excluded, thus the diff map threshold should exclude that pixel. The user must look at a lot of spectra from a lot of pixels to decide where, precisely, that threshold is best placed to retain all relevant information and discard noisy spectra. It may help to look at the $\chi^2$ as well. It doesn't help that the transient precursor phases are at the forming edge of each biomineral and have a less intense and noisier spectrum compared to the fully crystalline skeleton. Thus, human judgement is better than any automated system at this complex task.

9. If all the above is satisfactory, then they can export the MM as an RGB image, masked using the diff map with threshold, and unmasked individual pMaps, that is, grayscale images where the proportion ($0 < p < 1$) of each individual component is assigned a gray level between 0 and 255. The user then exports all relevant maps and images. These include the pMaps, the average image of the entire Ca acquisition that shows the morphology of the biomineral, the diff mask, and the $\chi^2$ mask, with $\chi^2 = 0.01$ as a threshold.

10. Finally, the user combines all images, maps, and masks in Adobe Photoshop 2023®, with each file in a separate layer in Photoshop. The same file contains all maps and masks for the repeat acquisition as well.

This processing was repeated on the same set of data by multiple people in parallel and multiple times by the same person, assessed both quantitatively and qualitatively. Quantitatively including $\chi^2$ values and percent error in identifying known phases, and qualitatively, for example, observing if the metastable phases appeared in places where they make sense and a visual inspection of the fit quality. With each repetition, each decision leading to improvement was held constant, eventually leading to the data processing presented here.

## The final Myriad Mapping (MM) of mesoscale carbonate-phases

Before all images, pMaps, and masks were stacked into layers of an Adobe Photoshop 2023® file, including duplicate and triplicate acquisitions, the pMaps were converted from grayscale images to duotone images in which white (grayscale level 255) in the original pMap (indicating 100% of the phase) was replaced with the color assigned to that phase, 0% was black, and every color in between represented intermediate concentrations. This left the quantitative information in each pMap unchanged. Then, the Magic Wand tool in Photoshop, with tolerance 128, was used to select a black pixel (0%) and all other pixels with <50% of concentration of that phase, in the duotone pMap. The

selected <50% pixels were deleted from that duotone pMap. In addition, any pixels that were masked off by any of the masking techniques described above were deleted. Both deletions were repeated for each of the 5 duotone pMap layers corresponding to each phase, thus the only pixels left in each pMaps were those with >50% of each phase.

This resulted in Myriad Maps (MMs) of mesoscale carbonate-phases, where >50% of each phase (ACCH$_2$O, ACC, CCHH, MHC, and aragonite or calcite, as presented in Figs. 2, 3, Supplementary Figs. 4–6, Supplementary Figs. 10–12. Duotone pMap in each Photoshop layer never overlapped because no pixel could ever have greater than 50% concentration of more than one phase. MMs were superimposed on a PEEM grayscale image, as shown in Figs. 2B, 3, Supplementary Fig. 10. The mature biomineral phase (aragonite or calcite) blue pixels were not displayed for clarity, so the skeletons remained visible. In Fig. 2A, Supplementary Figs. 4–6, 12, instead, the mature mineral phase was retained, as were the black masks. When superimposed on the PEEM image, MMs allowed the observer to see at once where all 4 precursor phases were localized, relative to the skeleton and tissue. Whereas including the mature biomineral gave a full picture of the composition of each pixel, especially when comparing with repeat acquisitions, e.g., in Supplementary Figs. 4–6.

Once all 86 areas were analyzed with MMs, including all images, maps, and masks in 86 separate Adobe Photoshop 2023® files, each including, in separate layers, all maps and masks for both the first and the repeat acquisitions (180 in total).

The data for each area were then summarized in Supplementary Data 2.xlsx.

Myriad means multiple, 5 in this work, but MM can be done with any number of phases, and it is used in Myriad Mapping to highlight the tremendous advantage of MM over any other kind of quantitative imaging, all of which are limited to 3 and only 3 phases, for example in RGB or CMY maps. This idea can easily be exported to a variety of other quantitative imaging and visualization methods, at any scale. Mesoscale is defined here as microscale field of view with nanoscale resolution. But the method could apply to any scale, from intergalactic to atomic scales.

## Synchrotron Infrared Nano Spectroscopy (SINS) Fourier Transform InfraRed (FTIR)

Two *Stylophora pistillata* coral samples, and synthetic carbonates were analyzed with SINS-FTIR. This method was selected after the failure of conventional attenuated total internal reflection–Fourier Transform InfraRed (ATR-FTIR) to detect any metastable precursor phases in freshly deposited coral skeleton surfaces. This result is not surprising because the probing depth of conventional FTIR methods is several microns into the sample surface. Thus, any surface phases on the coral skeleton are overwhelmed by the signal from crystalline aragonite, which was the only phase detected with ATR-FTIR.

SINS, instead, is surface-sensitive. It detects the FTIR signal from ~20 nm deep into the sample, and from a 20 nm wide region: a voxel with (20 nm)$^3$ volume. The depth sensitivity is an exponential decay, which to first order approximation, is equal to the tip radius, ~20 nm, but it is possible to sense things that are deeper depending on strength. In the present data, therefore, the stronger aragonite signal may originate from a bit deeper into the sample.

SINS-FTIR, on the other hand, is more surface sensitive than ATR-FTIR. Its sensitivity decays rapidly from the surface, thus its probing depth is typically on the order of the tip radius, in this case ~20 nm. Subsurface sampling is possible, depending on the material, tapping amplitude, and other experimental parameters, but it does not dominate the signal. The higher surface sensitivity and the higher lateral spatial resolution increase the ability of SINS-FTIR to differentiate heterogeneous surface phases from bulk aragonite.

SINS-FTIR data were collected at the Advanced Light Source, Lawrence Berkeley National Lab over multiple sessions using both

the Neaspec IR neaSCOPE instrument at Beamline 2.4 and a custom modified Bruker Innova AFM at Beamline 5.4. SINS was described in detail in previous work[73,74]. Briefly, broadband IR synchrotron radiation is introduced into an asymmetric Michelson interferometer in which one half of the incident light is focused onto the tip, oscillating in close proximity to the sample, and the other half is reflected off a moving flat mirror. The tapping oscillation frequency was between 250 and 300 kHz. The tips used were either platinum-coated silicon AFM tip (neaspec nano-FTIR tips, on beamline 2.4) or gold coated tips (PPP-NCHAu, on beamline 5.4). They both had resonant frequency 250–300 kHz. The light scattered by the near-field interaction of the tip with the sample interferes with the light reflected from the moving mirror and is detected by a mercury cadmium telluride (MCT) infrared detector. The signal is demodulated at higher harmonics of the tip-tapping frequency with a lock-in amplifier to remove far-field background contributions and then Fourier transformed to yield both an amplitude and a phase spectrum, which approximate the reflectance and absorbance spectrum, respectively[74]. Here, we display only 2nd harmonic amplitude data because the phase values were undefined in regions where the amplitude signal was near zero.

Although we chose to use SINS-FTIR because it is more surface sensitive than confocal FTIR or ATR-FTIR, its probing depth is still far greater than the 3 nm probed by PEEM at the Ca L-edge[35]. Another non-synchrotron-based FTIR method, Photo-induced Force Microscopy (PiFM) has 5 nm lateral and 20 nm depth resolution[26,62]. Because the surface layer of metastable phases, amorphous or crystalline, is just a few 100 s of nanometers, highly discontinuous, and interspersed with aragonite or calcite (Figs. 1–3, Supplementary Figs. 4–6, 10–12), the probability of observing either amorphous or CCHH precursors on the surface of fresh biominerals using SINS is much lower than in PEEM. After extensive analysis, however, we found a few CCHH-containing regions at the surface of 2 coral skeletons, presented in Fig. 4, Supplementary Fig. 7.

We observed peak splitting in CCHH and MHC, both synthetic and biogenic. We did not observe such splitting in any other regions of the skeleton other than near the forming surface. Peak splitting can, in principle, result from a technical artifact, observed when the AFM cantilever frequencies and the excitation source are out of sync and the intensities exceed the ideal range. Multiple lines of evidence suggest that the observed peak splitting is indeed characteristic of CCHH and MHC, and not an artifact. First, we observe precisely the same peak splitting in biogenic and synthetic CCHH and MHC. Second, we never observed it in synthetic ACC or aragonite, only in synthetic CCHH and MHC crystals. Third, in corals, we only observed peak splitting near the forming surface of coral skeletons. The observed peak splitting, therefore, cannot be artifact.

Other artifacts, including far-field effects and tip-sample coupling, can cause peak shifts. Far-field signals can be distinguished from near-field signals better at higher harmonics but at the cost of worse signal to noise ratio, thus, Bechtel et al. concluded that using second harmonic, as done in all of this work, was the best compromise[74]. Far-field effects, however, are expected to affect vast regions of the sample, not isolated 20 nm pixels, which is where CCHH and MHC were observed in this work.

SINS spectra of reference materials were obtained from powder samples, pressed onto carbon tape with a KBr press to provide a flat surface for AFM imaging (see Supplementary Fig. 8). Coral samples were embedded in epoxy and then polished. Polydimethylsiloxane (PDMS) often contaminates the AFM tips used for SINS. The amount of PDMS contamination varied from tip to tip, causing some spectra to have stronger PDMS peaks than others. The tip used during acquisition of the ACCH₂O, ACC, and aragonite spectra had a larger PDMS contamination, causing distinctive peaks especially around 825 and 1270 cm⁻¹. All CCHH and coral $v_3$ peaks were at different frequencies compared to PDMS peaks; thus, they were not affected by PDMS contamination. The peaks associated with PDMS are labeled by pink lines in Fig. 4.

We note that other IR nearfield AFM-based techniques besides SINS that have been used to probe calcium carbonate biominerals, including Photo-induced Force Microscopy (PiFM)[26,62].

### X-ray diffraction (XRD)
The mineralogical composition of each synthetically prepared $CaCO_3$ sample used in SINS analysis was assessed via XRD at the Lawrence Berkeley National Laboratory, Energy Geosciences Division. The instrumentation consists of a Rigaku SmartLab high-resolution X-ray diffractometer using Bragg-Brentano geometry. The diffractometer was equipped with a theta-theta goniometer and a rotating sample holder using a Cu cathode, with emission lines at $\lambda_{K\alpha1} = 1.5406$ Å and $\lambda_{K\alpha2} = 1.5444$ Å. Five-10 mg of fine, homogenized and flattened $CaCO_3$ powder was placed on a zero-background plate. Data from the sample powder were collected from 2θ values between 10° and 60°, with 0.02° steps, and count times of 2 s per step. The XRD data for CCHH and MHC are presented in Supplementary Fig. 9. All XRD data for all synthetic minerals are provided in file Supplementary Data 3.xlsx.

The composition of each synthetically prepared $CaCO_3$ sample used in PEEM analysis for the Cni16 component spectra was also assessed via XRD at the University of Wisconsin-Madison, Nanoscale Imaging and Analysis Center (NIAC). The instrumentation consists of a Bruker D8 Discovery X-ray diffractometer using a similar theta-theta goniometer and Cu cathode as described above. Five–10 mg of fine, homogenized and flattened $CaCO_3$ powder was placed on a glass slide. Data were collected from 2θ values at 20°, 40°, and 60°, and count times of 60 s per step. The integrated diffraction patterns from these acquisitions have a 2θ resolution of 0.005°. These data are provided in file Supplementary Data 3.xlsx.

### Reporting summary
Further information on research design is available in the Nature Portfolio Reporting Summary linked to this article.

## Data availability
All data are provided as supporting files. Stacks of raw images as acquired are available upon request to the corresponding author.

## Code availability
All code developed for this work, termed GG Macros, runs in Igor Pro 8 (Wavemetrics Inc.), and is downloadable free of charge on our website[70].

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

## Acknowledgements

We thank Tali Mass for reading the first draft of the manuscript, Philippe Ganot and Didier Zoccola for discussions, and Andreas Scholl for technical help during the PEEM experiments at ALS. Funding U.S. Department of Energy, Basic Energy Science, Chemical Sciences, Geosciences, Biosciences, Geosciences (DOE–BES–CSGB-Geosciences) Grants DE-FG02-07ER15899 and FWP-FP00011135 (PUPAG) U.S. National Science Foundation (NSF), Biomaterials Grant DMR-2220274 (PUPAG). U.S. National Science Foundation (NSF), Grant DMR–1720415 (UW-Madison Wisconsin Materials Research Science and Engineering Center, Nanoscale Imaging and Analysis Center (NIAC)). U.S. DOE Office of Science, Advanced Light Source, a User Facility under Contract no. DE-AC02-05CH11231 where all PEEM experiments were done.

## Author contributions

Conceptualization: P.U.P.A.G., C.A.S., Y.P., S.T., P.F., Methodology: C.A.S., E.T., C.A.St, Z.Z., M.A., C.C.A., L.S.D., P.U.P.A.G., Investigation: all co-authors, Data Processing and Visualization: S.A., C.P.B., C.L.F., A.H., C.N.K., I.J.K., I.M.L.C., J.M., M.P., A.P., V.Q., J.S.S., T.S., S.V. and E.J.W., Funding acquisition, administration, supervision: P.U.P.A.G., Writing—original draft: P.U.P.A.G., Writing—review and editing: all co-authors.

## Competing interests

The authors declare no competing interests.
