## [Peer Review File · Nature Communications]

Reviewers' Comments:

Reviewer #1:

Remarks to the Author:

This is very interesting and intriguing paper. Several potential artifacts are listed in appropriate section of the manuscript, however I have few additional questions that authors may address in the revised version:

(1) Preparation of samples involved "grinding and polishing [and use] (...) 22 g/L Na₂CO₃ in DI water (SI, p. 5). How the preparation procedures and the presence of DI water could influence crystallization and crystallization pathways of biogenic vs. abiotic amorphous phases? In "Potential artifacts" chapter the authors suggest that "even if more or faster crystallization occurred due to sample preparations, it is safe to assume that the amorphous phases were originally in intracellular vesicles, were then deposited by the living animal on the surface of the forming biominerals, and they crystallized into progressively more stable polymorphs". Would it be possible, that different crystallization pathways in molluscs/cnidarians vs. echinoderms result from selective removal (different solubility) of organic components in those taxa? In other words, it could be the selective dissolution/preparative modification of different for each group organic components that resulted in different crystallization pathways.

(2) The authors suggest that "both lines of evidence, temporal (Fig. S2-S4) and spatial (Fig. S9, 2, 3), suggest that CCHH and MHC occur as crystalline transient precursors to aragonitic but not to calcitic biominerals. This suggestion makes the process of biomineralization more complex than previously appreciated with four possible metastable phases, two amorphous and two crystalline". No explanation of this phenomenon is proposed, but the most plausible link would be to organic components (especially proteins) that are incorporated into biominerals. The authors mentioned only once the occurrence of biomineral proteins ("SM50 protein slows down the dehydration of ACC·2H₂O to ACC"), however, a brief suggestion re: possible causation of these different crystallization pathways would be useful (in addition to still another potential artifact mentioned in comment #1). It's a bit unfortunate, that the authors focused on nacre and not on bi-mineralic biogenic structures such as aragonitic(nacre)/calcitic(prismatic) mollusc shells. Such observations would instantly provide (or not) support to the main conclusion about different crystallization pathways leading to calcite and aragonite biominerals.

(3) Why only in coral and red abalone the identified mineral phases (ACC, CCHH, MHC) are present within more or less complete layer at the edge of the skeleton, whereas in echinoderm spine, the ACC is present only in few places outside, but also as aligned dots within the skeleton? It looks like the analysed skeletal region in echinoderm was not actively growing and the lack of other phases is an outcome of relatively late phase of disordered phase crystallization? Echinoderm stereom has layered structure thus not entirely understandable is a linear pattern visible in Fig. 3C with some spots identified as ACC.

(4) (page 2: line 28) "Ever since their [marine calcium carbonate biominerals] appearance in the Cambrian.." As far as I am aware, the first calcium carbonate biominerals appeared BEFORE Cambrian (Cloudina). Perhaps the authors may write "mass appearance"...

Reviewer #2:

Remarks to the Author:

Myriad Mapping (MM) mesoscale 1 mineral-phases reveals calcium carbonate hemihydrate in forming nacre and coral biominerals

This study, by Schmidt et al, re-investigates a collection of PEEM maps obtained from different biominerals (coral skeleton, nacre, and echinoderm spicules) for the potential presence of rarely reported calcium carbonate polymorphs. The two polymorphs monohydrocalcite (MHC) and calcium carbonate hemihydrate (CCHH) are of special interest here as they haven't been reported in natural biominerals. These phases are identified via better spectroscopic fit parameter results in a new data processing strategy referred to as Myriad Mapping. The authors identify CCHH and MHC

along the growth fronts of coral skeletons and nacre surfaces (which mature to aragonite) but only in insignificant amounts in the case of echinoderm spicules (which mature to calcite). This study furthers our understanding of biomineralization pathways and the energy landscapes underpinning the phase maturation processes in natural samples. I believe this is an interesting study for a broad range of readers as the field of biomineralization is intrinsically cross-disciplinary and believe this would be an exciting contribution to Nature Communications once the authors have addressed my questions and suggestions for improvement to the main text and supplementary.

Abstract

Page 1, lines 39-40: "Thus, biomineralization pathways are more complex than previously understood, ...". Would it be beneficial to change this to 'are more complex and diverse' to emphasize the differences observed between the coral skeleton and nacre versus sea urchin spines.

Introduction

Page 2, line 27: "These biominerals consistently outperform the sum of their parts (...)". The sentence appears to be missing a word, and it would be helpful to clarify what is being referred to here.

Results

Page 2, lines 36 to 41: I find the composition of the different sub-chapters in the results somewhat confusing as I think that a subchapter should be at least one paragraph and hence more than a single sentence (see first sub-chapter). This first subchapter in my view is superfluous as it only walks the reader through what the spectra not are, while the next subchapter investigates what they are. I suggest merging the first two results subchapters into one.

Figure 1: I suggest to re-arrange the panels (a and b) such that the spectra were displayed in a vertical instead of a horizontal layout. The figure will take up a little more space on a page but this would have the benefit of being able to insert vertical lines through the spectra marking the positions of the small peaks 4''' and 2, which would make it clearer for readers to compare them.

Page 3, line 33: "Myriad" highlights that this method displays quantitatively an unlimited number of phases. But this makes me wonder how many phases a person can perceive confidently (i.e., despite the range in brightness for each colour)? This is perhaps not unlimited? I suggest rephrasing this more realistically.

Figure 2 and 3 as well as (supplementary) Figure S2 to S4: The scalebars are bizarre. I can somewhat understand where the authors are coming from as they analyse the spectral the spectral information in each individual pixel. However, this does not imply that using a scale bar the size of the individual pixel is sufficient for the entire zoomed-out map! The resolution of a map is always determined by the diameter of the beam used to produce the maps (assuming the map was not binned or was under-sampled) this applies to any scanning probe techniques regardless of if they use electron beams, ion beams, or X-ray. This however cannot be the scale-bars size. The scale-bars are there to put the structural content of the maps into perspective. This means here that you would need to choose a scale bar size that allows readers to understand the size of the architectural features of the biominerals so they can see at one glance, e.g., how thick a nacre tablet is. As the map intends to show mesoscale features the scale bars need to be chosen appropriately. The size of the individual pixels could be added to the figure captions.

Discussion

Page 7, lines 37-38: "In all biominerals, the four precursor phases always mapped near surfaces, consistent with their being metastable." I don't understand this sentence please reword for clarity. Also, I think a single sentence cannot be a stand-alone paragraph. I suggest changing this to summarize the results to introduce your discussion.

Page 5, lines 24-25: The authors suggest that CCHH and MHC are transient precursors that mature to aragonitic and that this is not the case for calcite – I would like to encourage the authors to expand their thoughts a bit more in this direction: calcite and aragonite can also have different

trace elements (e.g., Mg²⁺) and organic constituents. Could the presence of CCHH and MHC be related to differences in trace elements and organic constituents between calcite and aragonite? Synthesising CCHH and MHC in the presence of the different organic phases may be an interesting outlook for future studies.

Page 10, lines 29 to 32: "If all phase transitions (solidification to ACCH₂O, dehydration, crystallization to CCHH, MHC, aragonite or calcite) occur as solid-state transformations (ref. 43), the composition of the final mineral must be governed by the first precursor phase." I see several problems with this statement that need to be discussed here: Firstly, how does a hydrous phase (ACCH₂O) transform to a crystalline phase by solid state? Obviously, this is possible if water was just absorbed to the ACC but in the case of ACCH₂O I believe this is thought to be present in throughout the phase (bulk)? The observations made here are also clearly at the growth front of the biominerals, which is in contact with fluids. Must not the presence of that much water argue for dissolution-reprecipitation? Secondly, a number of biominerals including nacre consist of nanogranules that are enveloped by an organic lining (see Jacob et al 2008, *Geochimica et Cosmochimica Acta* 72.22, which is not cited in the manuscript) – this lining may serve as a confined compartments for transformation via dissolution-reprecipitation. It may indeed be beneficial to expand a little on the different arguments for either transformation process at this stage of the manuscript. A recent study that is cited earlier in the manuscript but not in this section (Otter et al 2023, *Nature Communications*, 14, 1), argues for example for a transformation by dissolution-reprecipitation process. As these are perhaps the only recent studies investigating the phase transformation processes at high-resolution in natural biominerals. While this study investigates the complexity of transient phases that may stepwise transform at the growth front of the biominerals, Otter et al have observed chemically labelled vestiges of phase transformation processes present in the mature area behind the growth front but they did not discuss the possibility of that many transient phases. Would the authors consider the possibility that both transformation processes could be operating during different phases of the maturation sequence?

Methodology (Supplementary):

Page 3, 'Coral skeletons': I am missing a sentence explaining why the samples were treated with the used protocol, e.g., why the MgCl treatment (e.g., fixation)? In the case of fixation, was there any reason to assume they were still alive after a long transatlantic flight? What does Na Cacodylate powder do for your samples when used in combination with sodium carbonate? Also, I wonder if the authors could say something about why all the samples were chemically dried using the ethanol treatments (applicable also to nacre sample preparation)? What was the purpose of this? As ACC is known to transform following dehydration I wonder if any ACC was transformed into the transient crystalline phases observed here? I would like to invite the authors to talk a bit more about the reasons why they chose these sample preparation steps.

Page 3, 'Coral skeletons': "All synthetic crystals were embedded and polished as described above for coral skeletons." Change "above" to "below" as this is described in the following section.

Page 11, The final Myriad Mapping (MM) of mesoscale carbonate-phases resulting from the Cnidarian work: "But the method could apply to any scale, to intergalactic to atomic scales." Change "to" to "from" (i.e., "from intergalactic").

Page 12: "Here we chose to use SINS-FTIR because it is more surface sensitive than other IR methods such as confocal FTIR or ATR-FTIR." This is not true, 'Photo-induced Force Microscopy' (PiFM) uses metal-coated AFM tips and tunable laser to obtain spectra and maps at about 5 nm spatial resolution. Hence, I suggest changing this statement (i.e., "more surface sensitive than other IR methods").

Page 11: "...other kind of quantitative imaging, all of which are limited to 3 and only 3 phases, for example in RGB or CMY maps." I find this confusing, there exist enough quantitative maps that show more than three phases together in one map. Either this needs a lot more refinement as its not currently clear to me what the authors are trying to say here or the statement needs to be removed altogether.

Page 8, The Cnidarians: "... employing a group of extremely talented undergraduate students called "the Cnidarians", mostly from minorities underrepresented in science." This sounds like a fantastic

way to get the next generation of researchers engaged in a real and impactful science project and that they are included as co-authors. However, I find the way this is presented somewhat unfortunate. I acknowledge that diversity and inclusion is difficult to address in a way that sits well for everyone. However, I would recommend phrasing this part human-centric and not background-centric by providing the names and some details to each student involved including their background and involvement. In an academic context, recognizing the students by name and highlighting their specific roles and contributions can demonstrate a genuine commitment to diversity and inclusion. It also reinforces the idea that these students are valued participants in the research process rather than just tokens meant to represent diversity (which is how it can be currently understood and which is perhaps far from how it was intended).

Page 8, Decisions made by the Cnidarians: - "which are much better made by people than automatically or by any machine learning (ML) or artificial intelligence (AI) system." Please delete this part of the sentence as it is neither based on hard evidence nor references a paper backing the claim up.

Reviewer #3:

Remarks to the Author:

The study presented to the peer-review contains two main contributions: Firstly, the identification of two mineral precursor phases of biomineralization (CCHH and MHC). One of them (CCHH) had previously only been identified as a synthetic material. Secondly, previously recorded datasets of PEEM data have been reassessed for the presence of these phases, using an innovative approach of representing the data.

It is assumed that the work will be very significant to the field as a new pathway of biomineralization has been identified. As lined out by the authors, the study has an exploratory character, paving the way for other (crystallographic) techniques to be used in the characterization of biomineralization precursors. The study is based on a broad experience of the application of this technique on various samples containing biominerals.

Literature research did not yield any other studies that analyze CCHH and MHC as potential precursors for biomineralization.

Looking into the details of the claims, the evidence from PEEM seems rather well supported, whereas the corroborating nano-FTIR data is discussed rather superficially in the document, with main parts of the discussion moved into the SI.

In terms of the data analysis, some aspects are not very clear, and some major revision is deemed necessary. The same holds true for the description of the nano-FTIR experiments, that is lacking some essential information to reproduce the results. Part of the description of the analytical process is very technical and does not contain sufficient background on the mathematical and/ or physical assumptions. Furthermore, parts of the data analysis are based on subjective decisions and/or decisions that are not described in sufficient detail to reproduce the methodology.

This is very unfortunate, as the paper reveals some important results and certainly merits publication.

In the following, the more critical issues are listed:

p. 1 l. 37 The newly coined term Myriad mapping is introduced in the abstract, but not explained at this stage

p. 2 l. 37 The "Results" section is lacking some introductory word

p. 3 l. 16/17 The authors mention that the different phases of calcium carbonate are identified by the aspect ratios. However, the aspect ratio of the reference spectra and the sample are not shown and/or discussed

p. 3 Fig. 1 Whereas the identification in (A) is rather compelling, the distinction between CCHH and MHC in (B) is not so evident. The authors do not discuss the size of the phases and the possibility of mixed phases in a single pixel

p. 3 l. 34 The term mesoscale is slightly misleading, as the scales discussed here are microscale and nanoscale

p. 4 Fig. 2 The intensity distribution of the synthetic CCHH sample shows some additional peaks apart from $4''''$ – however these are not discussed

p. 5 Spatial distribution of phases: It would be interesting to see the distribution of the percentage of the CCHH and MHC contribution, also in the outermost layers of the aragonite-dominated area to confirm if there is a gradient in the concentration of these phases

p. 7 “Corroborating evidence from surface-sensitive infrared”:

The sample deposition is not discussed - what is the size of the sample and especially its thickness? Are the synthetic mineral phases studied as powders?

A figure with the experimental results should be included in the main text

p. 8 l. 16 The authors state that ACC was “detected” in 45%, 43% and 78% of all precursor pixels in coral, nacre, and sea urchin spines, respectively. However, it is not clear, whether this detection refers to the presence of a spectral contribution from ACC in the fit or if it refers to those pixels that show more than 50% of ACC

p. 11 The authors discuss several other, more accessible techniques that can be used to characterize similar samples, taking advantage of the crystalline properties of the biomineralization precursors. However, the more general implications of the new picture of the biomineralization process that has been obtained in this study, is not discussed, and should be added

Further to this, the extensive supporting information is lacking some information concerning the methodology. The following points should be addressed:

p. 3 The pressure for the pressurization in the EpoFix bath is missing

p. 4 For the description of the specimen and the corresponding experiments, a table could be easier to read

It is not clear, whether the regenerated sea urchin spines are cut a second time to obtain the specimen.

The abbreviation Cni16 is used here for the first time, but not explained.

p. 5 The coating of the samples for PEEM analysis is not explained

The sample preparation for the SINS-FTIR is lacking some information. Is the sample fixed on a substrate or are the slices used directly on the sample stage?

What is the thickness of the samples?

p. 6 The number of samples described in the SI seems to differ from the number of samples described in the main text

p. 7 The authors claim to have used the same polynomial background for all datasets in order to eliminate background effects. However, in doing so differences of the background (between measurement series and/or over time) are not considered. This seems only viable if the spectra have been taken under the same conditions

In terms of the peak fitting, it is not clear, why the “reasonable” polynomial background and arctangents for calcite were used for all samples. Also, it is not clear, how this reasonable background has been defined.

The manual adjustment of the symmetry of peaks 1 and 3 is not described. Is this a mathematical

transformation?

p. 8 Furthermore, the following aspects of the fitting routine should be verified:

Step 5) It is not clear, what the left and right cursor in the GG macro define

Step 6) It is not clear, whether this approach of "liberating" fit parameters one at a time can introduce a bias and can depend on the order of liberating parameters

Step 7) It seems that the constraint required here would be to assume the peak to be positive to avoid fitting the "dip" which should not require manual "trial and error"

Step 9) It seems that the same result could have been achieved by constraining the range of the peak position

p. 8/9 Whereas the project itself is very honourable and certainly merits a broad public propagation, I believe that the description of the methodology of the data analysis is not the right place.

The description should be limited to a more factual account.

There is no guarantee that a converging solution is always the right solution to a problem, especially as the participants involved in the data analysis are not necessarily working on their solutions fully independently.

For the scope of unbiased data analysis, an anonymous peer-based analysis seems a more robust approach.

The quality of the fit should additionally be compared using a reduced χ^2 to take the difference in degrees of freedom between a 3-, 4- or 5-component fit into account.

In terms of the 12 decisions that are to be made by the Cnidarians, some of them might be more objective if aided by computation. This is true for all criteria that are quantifiable, such as the alignment of images, the definition of noise level and the appropriate selection of the spectral region to be analyzed.

The description of the analysis process is very technical and lacks a description of the underlying mathematical process (e.g. what do the cursors do).

p. 11/12 The description of the SINS-FTIR approach is lacking some detail, especially concerning the tip and substrate used, the sample thickness and the experimental setup itself (commercial device?)

The authors should look into the penetration depth of the IR signal. The limitation to the penetration depth of 20 nm in all dimensions is not accurate. Whereas the lateral resolution is dominated by the tip size, the penetration depth essentially depends on the tapping amplitude of the cantilever as well as the harmonic that is chosen in the analysis. There are studies dedicated to obtaining some kind of tomography using SINS-FTIR. Furthermore, there is no consensus in literature concerning the penetration depth under standard conditions, with values reported varying from tens of nm to a couple of hundred nm.

Finally, some issues have been identified in the extended data section of the manuscript, which the authors should address:

S1) There is a huge difference between the number of single-pixel spectra recorded for e.g. 1-ACC (14) and 2-CCHH (20402). Is this difference taken into account when considering the single components for the fit? Presumably the spectrum of 1-ACC is much noisier than the one for 2-CCHH

S2) The improvement in the description when replacing the vaterite component with MHC or CCHH

raises the question, whether no vaterite is present or whether the model fails to distinguish between the two phases.

S3) According to the table the fit parameter of p_3 is 0. Probably the value is smaller than 0.0005 and a more appropriate representation should be chosen for the table.

As described in the caption, the area under the curve is represented, rather than the amplitude. The table should be adapted accordingly.

As discussed throughout the text, the main parameter to distinguish the components is the aspect ratio. These values should be included here.

For some of the peaks of the aragonite, all parameters are fixed. It is not clear how these parameters were obtained.

Fig. S1) The peak position of the 0-ACC-H₂O seems to be almost identical to 2-CCHH according to Table S3 and hence not distinct from all other carbonates.

Fig. S3) The pixel size is given as 19 nm, better than the best lateral resolution as described in the main text

Fig. S6) The figure contains an extensive caption of almost one page that would better fit into the main text or the description of the methodology.

The spectra shown presumably correspond to the absorption of the sample. In SINS-FTIR, the infrared absorption spectrum is typically obtained from the phase signal of the measurement. Some information on the computation of the spectra should be added.

It is not clear why different synthetic spectra are chosen for the comparison in S6A and S6B.

For a better link between the SINS spectra of the sample and the spectra of the synthetic minerals, a classical ATR-FTIR spectrum of the synthetic material should be obtained. This would be helpful for an unambiguous assignment.

The authors observe a significant shift of the peak frequencies in different directions (shift up to 50 cm^{-1} downwards and up to 65 cm^{-1} upwards). There is no explanation for this shift and in particular for the different shift for the sample described as "coral surface III" given.

Fig S9) The representation of experimental spectra of the sample interspersed with spectra of the synthetic material is not very clear. The synthetic spectra should be highlighted (e.g. by dotted lines)

Reviewer #1 (Remarks to the Author):

This is very interesting and intriguing paper.

Thank you! We appreciate the positive feedback.

Several potential artifacts are listed in appropriate section of the manuscript, however I have few additional questions that authors may address in the revised version:

(1) Preparation of samples involved “grinding and polishing [and use] (...) 22 g/L Na₂CO₃ in DI water (SI, p. 5). How the preparation procedures and the presence of DI water could influence crystallization and crystallization pathways of biogenic vs. abiotic amorphous phases?

On page 4 of the SI we now explained “It is important to prevent dissolution or (re)crystallization of any amorphous or metastable crystalline precursor phases in coral skeletons during sample preparation. Thus, all sawing, grinding, and polishing solutions contained a concentration of -CO₃ much greater than the dissolution concentration of ACC. These are, respectively, 22 g/L and 6 mg/L. No Ca was ever added to any of these solutions, thus, all the Ca analyzed in biominerals in this work was originally present in the biominerals.”

In “Potential artifacts” chapter the authors suggest that “even if more or faster crystallization occurred due to sample preparations, it is safe to assume that the amorphous phases were originally in intracellular vesicles, were then deposited by the living animal on the surface of the forming biominerals, and they crystallized into progressively more stable polymorphs”. Would it be possible, that different crystallization pathways in molluscs/cnidarians vs. echinoderms result from selective removal (different solubility) of organic components in those taxa? In other words, it could be the selective dissolution/preparative modification of different for each group organic components that resulted in different crystallization pathways.

We thank the reviewer for this idea, which is certainly an alternative potential artifact, thus we added a comment to this effect on page 15: “Another potential artifact is differential dissolution of aragonite vs. calcite biominerals due to varying organic molecules. These are certainly different in each organism, and they affect solubility and therefore potential dissolution artifacts differently in each biomineral.”

(2) The authors suggest that “both lines of evidence, temporal (Fig. S2-S4) and spatial (Fig. S9, 2, 3), suggest that CCHH and MHC occur as crystalline transient precursors to aragonitic but not to calcitic biominerals. This suggestion makes the process of biomineralization more complex than previously appreciated with four possible metastable phases, two amorphous and two crystalline”. No explanation of this phenomenon is proposed, but the most plausible link would be to organic components (especially proteins) that are incorporated into biominerals. The authors mentioned only once the occurrence of biomineral proteins (“SM50 protein slows down the dehydration

of ACC_{H2O} to ACC”), however, a brief suggestion re: possible causation of these different crystallization pathways would be useful (in addition to still another potential artifact mentioned in comment #1). It’s a bit unfortunate, that the authors focused on nacre and not on bi-mineralic biogenic structures such as aragonitic(nacre)/calcitic(prismatic) mollusc shells. Such observations would instantly provide (or not) support to the main conclusion about different crystallization pathways leading to calcite and aragonite biominerals.

What a great idea! Thank you! We now suggest it in the conclusions on page 16 “One important experiment in the future will be the observation of aragonite and calcite biominerals formed simultaneously by the same organism, for example mollusk shell’s nacre aragonite and prismatic calcite at the forming lip of either bivalve or gastropod shells. If these experiments demonstrate that CCHH and MHC precursors exist in aragonite but not calcite forming biominerals, that will provide even stronger and less artifact-prone evidence than presented here. Other interesting experiments could explore the effect of proteins or additives (e.g. Mg) on selection of the different pathways to aragonite or calcite.”

We also mention the potential contribution of proteins to the different pathways on page 13 “This control, and indeed the existence of the different pathways in Fig. 5, may be mediated by organic molecules, especially proteins.”.

(3) Why only in coral and red abalone the identified mineral phases (ACC, CCHH, MHC) are present within more or less complete layer at the edge of the skeleton, whereas in echinoderm spine, the ACC is present only in few places outside, but also as aligned dots within the skeleton? It looks like the analysed skeletal region in echinoderm was not actively growing and the lack of other phases is an outcome of relatively late phase of disordered phase crystallization? Echinoderm stereom has layered structure thus not entirely understandable is a linear pattern visible in Fig. 3C with some spots identified as ACC.

We thank the reviewer for this astute observation. One important decision to make when processing MM data is where to place the boundary between the skeleton and the background, in other words, where the edge of the biomineral is, which we term “masking”. This is described at length in the SI (pages 11,12), and SI figure Fig. S4 shows that the green pixels are the surface, as defined by the black mask, thus, they line up at the edge of the spines not inside of them. Spectra from the pixels just outside the spines surface are much less intense, noisier, and therefore the χ^2 of the fit is so poor that we excluded them. Of course, we agree that the stereom is layered (and Fig. 3C1 shows it), but we don’t observe a change in mineralogy, therefore all layers inside the stereom bulk appear as blue calcite.

(4) (page 2: line 28) “Ever since their [marine calcium carbonate biominerals] appearance in the Cambrian..” As far as I am aware, the first calcium carbonate biominerals appeared BEFORE Cambrian (Cloudina). Perhaps the authors may write “mass appearance”...

Thank you. We changed “appearance” to “proliferation”.

Reviewer #2 (Remarks to the Author):

Myriad Mapping (MM) mesoscale 1 mineral-phases reveals calcium carbonate hemihydrate in forming nacre and coral biominerals

This study, by Schmidt et al, re-investigates a collection of PEEM maps obtained from different biominerals (coral skeleton, nacre, and echinoderm spicules) for the potential presence of rarely reported calcium carbonate polymorphs. The two polymorphs monohydrocalcite (MHC) and calcium carbonate hemihydrate (CCHH) are of special interest here as they haven't been reported in natural biominerals. These phases are identified via better spectroscopic fit parameter results in a new data processing strategy referred to as Myriad Mapping. The authors identify CCHH and MHC along the growth fronts of coral skeletons and nacre surfaces (which mature to aragonite) but only in insignificant amounts in the case of echinoderm spicules (which mature to calcite). This study furthers our understanding of biomineralization pathways and the energy landscapes underpinning the phase maturation processes in natural samples. I believe this is an interesting study for a broad range of readers as the field of biomineralization is intrinsically cross-disciplinary and believe this would be an exciting contribution to Nature Communications once the authors have addressed my questions and suggestions for improvement to the main text and supplementary.

Thank you for the positive feedback!

Abstract

Page 1, lines 39-40: "Thus, biomineralization pathways are more complex than previously understood, ...". Would it be beneficial to change this to 'are more complex and diverse' to emphasize the differences observed between the coral skeleton and nacre versus sea urchin spines.

We added "and diverse" to the abstract. **Page 1.**

Introduction

Page 2, line 27: "These biominerals consistently outperform the sum of their parts (...)". The sentence appears to be missing a word, and it would be helpful to clarify what is being referred to here.

We changed it to: "These biominerals consistently outperform the sum of their components, inspiring strategies to design novel materials and metamaterials". See **page 2.**

Results

Page 2, lines 36 to 41: I find the composition of the different sub-chapters in the results somewhat confusing as I think that a subchapter should be at least one paragraph and hence more than a single sentence (see first sub-chapter). This first subchapter in my view is superfluous as it only walks the reader through what the spectra not are, while the next subchapter investigates what they are. I suggest merging the first two results subchapters into one.

We agree that the first subchapter was too short, so we explained better what we meant in multiple sentences. We retain it to highlight how the mystery phases came up. See “Unknown CaCO₃ mineral phases” page 3.

Figure 1: I suggest to re-arrange the panels (a and b) such that the spectra were displayed in a vertical instead of a horizontal layout. The figure will take up a little more space on a page but this would have the benefit of being able to insert vertical lines through the spectra marking the positions of the small peaks 4” and 2, which would make it clearer for readers to compare them.

This was a good suggestion, and we made precisely the change recommended. See revised Fig. 1.

Page 3, line 33: “Myriad” highlights that this method displays quantitatively an unlimited number of phases. But this makes me wonder how many phases a person can perceive confidently (i.e., despite the range in brightness for each colour)? This is perhaps not unlimited? I suggest rephrasing this more realistically.

We inserted a comment on page 5 to this effect: “For human observation at a glance, a dozen phases can easily be distinguished.”

Figure 2 and 3 as well as (supplementary) Figure S2 to S4: The scalebars are bizarre. I can somewhat understand where the authors are coming from as they analyse the spectral the spectral information in each individual pixel. However, this does not imply that using a scale bar the size of the individual pixel is sufficient for the entire zoomed-out map! The resolution of a map is always determined by the diameter of the beam used to produce the maps (assuming the map was not binned or was under-sampled) this applies to any scanning probe techniques regardless of if they use electron beams, ion beams, or X-ray. This however cannot be the scale-bars size. The scale-bars are there to put the structural content of the maps into perspective. This means here that you would need to choose a scale bar size that allows readers to understand the size of the architectural features of the biominerals so they can see at one glance, e.g., how thick a nacre tablet is. As the map intends to show mesoscale features the scale bars need to be chosen appropriately. The size of the individual pixels could be added to the figure captions.

We agree with the reviewer that for scanning probe microscopies of all kinds the pixels size and the beam size cannot be the same. However, the method used here is PEEM, a full-field microscopy, not a scanning probe microscopy method. The resolution of the method is 20 nm, but we used lower magnification thus the pixels sizes (19, 57, 60 nm) really are the resolution of the images. In the case of the 19 nm pixels, it simply means the single pixel spectra are oversampled by ~0.5 nm on each side. We agree with the reviewer that the scalebars as presented were not informative on the biomineral size, only on the MM pixel size, thus, we have added both types of scalebars to each figure. See revised Figs. 2, 3, S4-S6, S10-S12.

Discussion

Page 7, lines 37-38: “In all biominerals, the four precursor phases always mapped near surfaces, consistent with their being metastable.” I don’t understand this sentence please reword for clarity. Also, I think a single sentence cannot be a stand-alone paragraph. I suggest changing this to summarize the results to introduce your discussion.

Thank you, we rephrased it to: “In all biominerals, the four metastable phases always mapped near the forming surfaces, consistent with their being precursors to biomineral formation.” We also joined this sentence with the next paragraph. See page 11.

Page 5, lines 24-25: The authors suggest that CCHH and MHC are transient precursors that mature to aragonitic and that this is not the case for calcite – I would like to encourage the authors to expand their thoughts a bit more in this direction: calcite and aragonite can also have different trace elements (e.g., Mg²⁺) and organic constituents. Could the presence of CCHH and MHC be related to differences in trace elements and organic constituents between calcite and aragonite? Synthesising CCHH and MHC in the presence of the different organic phases may be an interesting outlook for future studies.

We cannot say more based on the current data, but we added a comment on page 16: “Other interesting experiments could explore the effect of proteins or additives (e.g. Mg) on selection of the different pathways to aragonite or calcite.”

We also added a comment to this effect on page 13: “This control, and indeed the existence of the different pathways in Fig. 5, may be mediated by organic molecules, especially proteins.”

Page 10, lines 29 to 32: “If all phase transitions (solidification to ACCH₂O, dehydration, crystallization to CCHH, MHC, aragonite or calcite) occur as solid-state transformations (ref. 43), the composition of the final mineral must be governed by the first precursor phase.” I see several problems with this statement that need to be discussed here: Firstly, how does a hydrous phase (ACCH₂O) transform to a crystalline phase by solid state? Obviously, this is possible if water was just absorbed to the ACC but in the case of ACCH₂O I believe this is thought to be present in throughout the phase (bulk)?

We honestly do not know. All we can say for sure is that the spectra we detect look like the phases we assign them to. We also see them transitioning from one movie to the next, once the biomineral is removed from the animal, dehydrated, and placed in ultra-high vacuum. Of course, this is not in any way representative of what happens in nature, but it helps clarify that a certain precursor phase is indeed a precursor. We leave the possibility of dissolution and reprecipitation entirely open, as discussed in a new discussion heading on “Solid-state transformation or dissolution and reprecipitation?”. See pages 14-15.

The observations made here are also clearly at the growth front of the biominerals, which is in contact with fluids. Must not the presence of that much water argue for dissolution-reprecipitation?

We now did a much better job at discussing the two possible mechanisms in a new discussion heading on “Solid-state transformation or dissolution and reprecipitation?”. See pages 14-15, where we say: “Since all precursor phases, amorphous and metastable crystalline, were always observed at the forming surface of all biominerals, phase transitions certainly have access to water, thus, it is entirely possible that dissolution and reprecipitation occur, especially if the saturation was close to $W_{\text{arag or calcite}} \cong 1$, or that of seawater, $\Omega_{\text{arag}} \cong 3$. In corals, however, in the extracellular calcifying fluid the supersaturation was measured with micro-electrodes to be $\Omega_{\text{arag}} \cong 12$ ^{24, 48}. Such supersaturation allows for the dissolution of precursor carbonates but not aragonite dissolution. Even in corals, therefore, dissolution and reprecipitation may be the dominant mechanism for aragonite formation. However, if this was the only mechanism, in the solid skeleton the metastable precursor phases observed here, amorphous and crystalline, would never be observe. Their observation implies unequivocally that at least some precursor particles never dissolve and reprecipitate, but undergo solid-state transformation.

Sun et al. argued that in corals both mechanisms must occur based on the porosity of the surface layer observed in PEEM experiments and the space filling nature of all mature coral skeletons²⁴. In nacre, instead, Otter et al.²⁶ showed that the dissolution and reprecipitation mechanism dominates. Our data certainly do not contradict theirs. Our observation of metastable precursors at the forming surface of nacre may be interpreted in two different ways. In one scenario, some precursor particles dissolve more slowly than others, and those slow-precursors are observed at the surface. They eventually dissolve and reprecipitate, thus, they never undergo a solid-state transformation. In another scenario, the precursor particles never dissolve but undergo a solid-state transformation. We do know that it is possible to crystallize synthetic ACCH₂O into calcite via a solid-state transformation, as detected by NMR spectroscopy⁴⁹.

One could speculate that perhaps both mechanisms could occur in various biominerals, but to different extents, or at different locations or formation stages. Further studies, designed to tease these two mechanisms apart, will shed light. The present study was not designed to address these mechanisms thus, it cannot settle the debate. ”

Secondly, a number of biominerals including nacre consist of nanogranules that are enveloped by an organic lining (see Jacob et al 2008, *Geochimica et Cosmochimica Acta* 72.22, which is not cited in the manuscript) – this lining may serve as a confined compartments for transformation via dissolution-reprecipitation. It may indeed be beneficial to expand a little on the different arguments for either transformation process at this stage of the manuscript. A recent study that is cited earlier in the manuscript but not in this section (Otter et al 2023, *Nature Communications*, 14, 1), argues for example for a transformation by dissolution-reprecipitation process. As these are perhaps the

only recent studies investigating the phase transformation processes at high-resolution in natural biominerals. While this study investigates the complexity of transient phases that may stepwise transform at the growth front of the biominerals, Otter et al have observed chemically labelled vestiges of phase transformation processes present in the mature area behind the growth front but they did not discuss the possibility of that many transient phases. Would the authors consider the possibility that both transformation processes could be operating during different phases of the maturation sequence?

Yes, as mentioned above, both mechanisms may occur simultaneously, or, as the reviewer suggests at different maturation stages. I think someone, perhaps the reviewer, could design a study to discriminate the two mechanisms.

Methodology (Supplementary):

Page 3, 'Coral skeletons': I am missing a sentence explaining why the samples were treated with the used protocol, e.g., why the MgCl treatment (e.g., fixation)?

On SI page 4 we now wrote "All corals were first immersed in seawater enriched with 5 w% MgCl₂ to relax the tissues, and prevent the retraction reaction that polyps exhibit when touched."

In the case of fixation, was there any reason to assume they were still alive after a long transatlantic flight?

No, fixation was done right away in Monaco, as was every other step of the preparation until embedding, as explained in the methods (SI page 3, "• shipped by courier to either Madison, WI or Berkeley, CA." Immediately before beamtime in Berkeley, we cut open the blocks of epoxy, ground, and polished surfaces and analyzed corals.

What does Na Cacodylate powder do for your samples when used in combination with sodium carbonate?

SI page 4: "The cacodylate buffer was used during fixation in paraformaldehyde because it helped better preserve the tissue, which remained attached to the biomineral growth front, and therefore protected the delicate metastable precursor phases at the biomineral surface."

Also, I wonder if the authors could say something about why all the samples were chemically dried using the ethanol treatments (applicable also to nacre sample preparation)? What was the purpose of this? As ACC is known to transform following dehydration I wonder if any ACC was transformed into the transient crystalline phases observed here? I would like to invite the authors to talk a bit more about the reasons why they chose these sample preparation steps.

See SI page 4: "Corals and all other samples described here were dehydrated in ethanol because ethanol binds to ACC and all other carbonates¹, therefore protecting metastable phases from transforming^{2,3,4,5}. A protective layer of ethanol before

embedding better preserves fresh samples. Right before beamtime, the polished surface is also soaked with ethanol, so the metastable precursors are stabilized by a surface monolayer of ethanol.”

Page 3, ‘Coral skeletons’: “All synthetic crystals were embedded and polished as described above for coral skeletons.” Change “above” to “below” as this is described in the following section.

Corrected it on SI page 5, in 2 places.

Page 11, The final Myriad Mapping (MM) of mesoscale carbonate-phases resulting from the Cnidarian work: “But the method could apply to any scale, to intergalactic to atomic scales.” Change “to” to “from” (i.e., “from intergalactic”).

Changed “to” to “from” on SI page 13.

Page 12: “Here we chose to use SINS-FTIR because it is more surface sensitive than other IR methods such as confocal FTIR or ATR-FTIR.” This is not true, ‘Photo-induced Force Microscopy’ (PiFM) uses metal-coated AFM tips and tunable laser to obtain spectra and maps at about 5 nm spatial resolution. Hence, I suggest changing this statement (i.e., “more surface sensitive than other IR methods”).

Thanks for this great suggestion! We added on SI page 14: “Another non-synchrotron-based FTIR method, Photo-induced Force Microscopy (PiFM) has 5 nm lateral and depth resolution¹⁸.” We also listed PiFM among the promising methods to reveal CCHH MHC in the conclusions. Page 16 of the main text: “Photo-induced Force Microscopy (PiFM) with 5 nm lateral resolution and 20 nm depth sensitivity is also sensitive to carbonate FTIR spectroscopy⁵². Importantly, EBSD, nanoSIMS, and PiFM are commercial instruments vastly more widely available than the synchrotron.....”

Page 11: “...other kind of quantitative imaging, all of which are limited to 3 and only 3 phases, for example in RGB or CMY maps.” I find this confusing, there exist enough quantitative maps that show more than three phases together in one map. Either this needs a lot more refinement as its not currently clear to me what the authors are trying to say here or the statement needs to be removed altogether.

We understand what the reviewer means: if one looks at a photograph of a granite one can see many phases all distinguished by different colors. However, a photograph of granite provides non-quantitative information. Quantitative maps, e.g. EBSD maps of crystal orientations are limited to 3 primary orientations and colors at a time, e.g. CMY, displaying intermediate orientations with intermediate colors. Another example of quantitative mapping is obtained with x-ray fluorescence of different elements, or component mapping of different phases, and these are always limited to 3 and only 3 phases, each associated to a primary color (usually RGB), with mixtures of phases having intermediate colors. Three and only 3 phases can be displayed at once, if one wants to retain quantitative information of mixed phases. With MM we choose to display

more phases, but only the majority phase (>50%) is displayed in each pixel, and information on mixed phase pixels is lost.

To make this point even clearer, if we maintained fully quantitative information also <50%, a yellow pixel could result from the additive mixture of 50% red and 50% green phases, or from the 100% yellow phase. Such a map cannot be quantitative and unambiguous. This is why mixed phases have to go.

Page 8, The Cnidarians: "... employing a group of extremely talented undergraduate students called "the Cnidarians", mostly from minorities underrepresented in science." This sounds like a fantastic way to get the next generation of researchers engaged in a real and impactful science project and that they are included as co-authors. However, I find the way this is presented somewhat unfortunate. I acknowledge that diversity and inclusion is difficult to address in a way that sits well for everyone. However, I would recommend phrasing this part human-centric and not background-centric by providing the names and some details to each student involved including their background and involvement. In an academic context, recognizing the students by name and highlighting their specific roles and contributions can demonstrate a genuine commitment to diversity and inclusion. It also reinforces the idea that these students are valued participants in the research process rather than just tokens meant to represent diversity (which is how it can be currently understood and which is perhaps far from how it was intended).

We removed any mention of the Cnidarians.

Page 8, Decisions made by the Cnidarians: - "which are much better made by people than automatically or by any machine learning (ML) or artificial intelligence (AI) system." Please delete this part of the sentence as it is neither based on hard evidence nor references a paper backing the claim up.

It is based on hard evidence and numbers, but we had not previously provided them. We now wrote on **pages 9-10 of the SI**: "We estimate that automating these decisions, thus, rigorously exploring the entire parameter space for each decision, will take 2-4 years of continuous operation for a 10 TFLOPS (tera floating point operations) computer (e.g. an Apple M2 Max CPU, or an AMD Ryzen 5, both with ~3.5 GHz, and 10-100 GB of RAM)."

Reviewer #3 (Remarks to the Author):

The study presented to the peer-review contains two main contributions: Firstly, the identification of two mineral precursor phases of biomineralization (CCHH and MHC). One of them (CCHH) had previously only been identified as a synthetic material. Secondly, previously recorded datasets of PEEM data have been reassessed for the presence of these phases, using an innovative approach of representing the data.

It is assumed that the work will be very significant to the field as a new pathway of biomineralization has been identified. As lined out by the authors, the study has an exploratory character, paving the way for other (crystallographic) techniques to be used in the characterization of biomineralization precursors. The study is based on a broad experience of the application of this technique on various samples containing biominerals.

Literature research did not yield any other studies that analyze CCHH and MHC as potential precursors for biomineralization.

Looking into the details of the claims, the evidence from PEEM seems rather well supported,

Thank you for the positive feedback.

whereas the corroborating nano-FTIR data is discussed rather superficially in the document, with main parts of the discussion moved into the SI.

We thank the reviewer for this criticism. We moved the main FTIR figure to the main text, expanded its description in the main text. See pages 9-11.

In terms of the data analysis, some aspects are not very clear, and some major revision is deemed necessary. The same holds true for the description of the nano-FTIR experiments, that is lacking some essential information to reproduce the results. Part of the description of the analytical process is very technical and does not contain sufficient background on the mathematical and/ or physical assumptions. Furthermore, parts of the data analysis are based on subjective decisions and/or decisions that are not described in sufficient detail to reproduce the methodology.

This is very unfortunate, as the paper reveals some important results and certainly merits publication.

Thank you for saying that the paper is worth publishing, and for identifying where it could be improved. We really appreciate all the time and effort your expert suggestions took, and we agree that the improvements you suggest make the paper stronger.

In the following, the more critical issues are listed:

p. 1 l. 37 The newly coined term Myriad mapping is introduced in the abstract, but not explained at this stage

Sorry, but there is a strict word limit at 150 words, so it is impossible to explain it there.

p. 2 l. 37 The “Results” section is lacking some introductory word

We added an introductory paragraph saying on page 2: “Below we describe how we observed two unknown CaCO₃ mineral phases using soft-x-ray spectroscopy at the nanoscale, how we identified them as CCHH and MHC, and therefore needed to develop a new method that allows to display quantitatively more phases. Using the new method we observed the spatial distribution of 5 distinct phases and their occurrence across diverse biominerals. We also provide corroborating evidence from a completely independent infrared method.”

p. 3 l. 16/17 The authors mention that the different phases of calcium carbonate are identified by the aspect ratios. However, the aspect ratio of the reference spectra and the sample are not shown and/or discussed

We thank the reviewer for pointing out this deficiency. We have completely rearranged Fig. 1 so that the unknown spectra overlap the component spectra of the known mineral phases. This shows that the relative intensities of the main peaks are very different from one phase to another.

p. 3 Fig. 1 Whereas the identification in (A) is rather compelling, the distinction between CCHH and MHC in (B) is not so evident. The authors do not discuss the size of the phases and the possibility of mixed phases in a single pixel

Again thanks for the great criticism. With the new arrangement Fig. 1 both unknown spectra in A and B are more evidently matched or mismatched with the known phases. Or so we hope. We also added the percentage assigned to each pixel in 5 component analysis to the Fig. 1 caption on page 4.

p. 3 l. 34 The term mesoscale is slightly misleading, as the scales discussed here are microscale and nanoscale

The reviewer is right, thus we changed to “nanoscale” in the title and in the text, because that’s the scale at which the MM pixels are displayed. On page 5 we wrote: “All MMs were acquired with microscale field of view and nanometer resolution, but MM can be used for quantitative imaging at intergalactic or atomic scales.”

p. 4 Fig. 2 The intensity distribution of the synthetic CCHH sample shows some additional peaks apart from 4” – however these are not discussed

We wrote on page 5: “The CCHH and MHC characteristic peak positions are highlighted by gray vertical lines in Fig. 1, so the reader can observe the shifts in energy of relevant

peaks in all other known spectra compared to CCHH and MHC. The CCHH peak positions are indicated by cyan vertical lines in Fig. S1CD. The amplitudes of all peaks and therefore the peak intensities are also clearly different across the known spectra. In Fig. S1CD, peak positions and intensities can directly be compared across phases: aragonite has the lowest intensity peak 2, calcite has the highest, the other have intermediate intensities but their energy positions vary (Fig. S1D). MHC has the highest peak 4 and aragonite has the lowest, the other phases have intermediate intensities (Fig. S1C). The smaller crystalline peaks labelled 4', 4'', 4''' in Figs. 1 and S1 are too low in intensity to be distinguished from the noise in single-pixel spectra, but stand out when averaging more than 20 pixels. Even though they can't be distinguished, the lineshape of the noisy spectra in this region is distinct from the valley that exists in this region in ACC and ACCH₂O spectra (Fig. 1)."

We wrote on page 6: "The fit of unknown spectra with known component spectra included all peak parameters: position, amplitude, and width for each known phase, thus each fit took into account all peaks and their peak parameters."

p. 5 Spatial distribution of phases: It would be interesting to see the distribution of the percentage of the CCHH and MHC contribution, also in the outermost layers of the aragonite-dominated area to confirm if there is a gradient in the concentration of these phases

We did show in Fig. S11 (formerly S9) the gradual transition from ACCH₂O to CCHH to aragonite. However, the reviewer is asking for more quantitative information about the remainder phases, not the dominant >50% phases. Thus, we added a new Fig. S12 where we show the same data as in Figs. 2 and S11 but displayed as RGB and CMY maps so that the mixed phases can be assessed quantitatively.

p. 7 "Corroborating evidence from surface-sensitive infrared":
The sample deposition is not discussed - what is the size of the sample and especially its thickness? Are the synthetic mineral phases studied as powders?
A figure with the experimental results should be included in the main text

We moved the SINS-FTIR data to the main text, and we expanded their discussion. See pages 9-11. We also expanded the methods in the SI to include a description of the powders, and their compression with a KBr press, and added photos of samples and KBr press into new Fig. S8.

p. 8 l. 16 The authors state that ACC was "detected" in 45%, 43% and 78% of all precursor pixels in coral, nacre, and sea urchin spines, respectively. However, it is not clear, whether this detection refers to the presence of a spectral contribution from ACC in the fit or if it refers to those pixels that show more than 50% of ACC

Great point. Thank you! We now wrote on page 11: "...in nacre and coral, observed at >50% concentration in 48% and 46% of precursor pixels, respectively."

p. 11 The authors discuss several other, more accessible techniques that can be used to characterize similar samples, taking advantage of the crystalline properties of the biomineralization precursors. However, the more general implications of the new picture of the biomineralization process that has been obtained in this study, is not discussed, and should be added

We had discussed it to the extent that we can. If the reviewer has a more specific request, we will gladly satisfy it. Specifically, we provide a completely new energy landscape in Fig. 5, and a comment about biological control on page 13: “During biomineralization, the more complex the energy landscape, the greater the number of phase transitions necessary to reach the final stable polymorph, the more biological control the organism has over the crystallization process. This control, and indeed the existence of the different pathways in Fig. 5, may be mediated by organic molecules, especially proteins. In sea urchin biominerals, for example, the SM50 protein slows down the dehydration of $ACCH_2O$ to ACC²¹. Catalysis is also possible, rather than inhibition of phase transitions³⁷. A complicated system of phase transitions may provide a benefit to organisms: by gaining control over the crystallization process, organisms more effectively direct it towards the crystal polymorph, morphology³⁸, size, and crystal arrangements³⁹ that provide them with better function and therefore greater evolutionary advantage.”.

Further to this, the extensive supporting information is lacking some information concerning the methodology. The following points should be addressed:

p. 3 The pressure for the pressurization in the EpoFix bath is missing

The pressure was missing from one of the bullets and appeared in the following one, we now mentioned it the first time “pressurizing” appeared. See SI page 3: “immersed in two EpoFix baths before pressurizing them to 2 atm in a third EpoFix bath.”

p. 4 For the description of the specimen and the corresponding experiments, a table could be easier to read. It is not clear, whether the regenerated sea urchin spines are cut a second time to obtain the specimen.

We now say on SI page 5: “After regeneration, entire spines were fixed, dehydrated, the most mature part of each spine was cut so that the regenerating parts were retained and 4 spines fit into 1-inch round molds, and embedded as described above and in a previous paper⁹. Figure 2 in Albéric et al. 2019 shows the regenerating and mature parts of 4 spines after embedding and polishing⁹.”

The abbreviation Cni16 is used here for the first time, but not explained.

We removed it, as it was unnecessary there.

p. 5 The coating of the samples for PEEM analysis is not explained.

On SI page 6 we now say: “All samples for PEEM analysis were coated with 1nm Pt in the area of interest and 40 nm Pt all around it¹², to make the electrical contact necessary for the sample to be at high voltage for analysis. This was achieved using a high resolution Cressington 208HR (Cressington, UK, Ted Pella, USA) sputter coater with a rotary and planetary tilt stage. The area of interest was covered by a diced Si wafer 4 mm x 4 mm, the surrounding area was coated with 40 nm Pt with the sample static and not tilted. The chamber was vented, the Si wafer removed, then the sample was coated with 1 nm Pt while tilted and spinning at the max speed possible to guarantee homogeneity of the coated thin 1nm layer of Pt.^{11, 12, 13}”

The sample preparation for the SINS-FTIR is lacking some information. Is the sample fixed on a substrate or are the slices used directly on the sample stage? What is the thickness of the samples?

We have now described in much more detail how the samples for SINS-FTIR were mounted and prepared, and we inserted a new Fig. S8, including the first observation of blue ACCH₂O! this is also discussed on SI page 5.

p. 6 The number of samples described in the SI seems to differ from the number of samples described in the main text

We apologize, there were only 54 coral samples not 56, thus we corrected it in the SI. Thanks for catching this typo.

p. 7 The authors claim to have used the same polynomial background for all datasets in order to eliminate background effects. However, in doing so differences of the background (between measurement series and/or over time) are not considered. This seems only viable if the spectra have been taken under the same conditions.

We now explain on SI page 8: “All spectra were acquired under identical conditions and on the same beamline. The beamline background intensity is completely flat and featureless at the Ca L-edge energies, thus, this choice is justified.”

In terms of the peak fitting, it is not clear, why the “reasonable” polynomial background and arctangents for calcite were used for all samples. Also, it is not clear, how this reasonable background has been defined.

Thank you. We now explained on SI page 8: “Arctangents for calcite were found, placed 0.25 eV below peak 3 and below peak 1, fixed in position, width, and amplitude, and re-used identically for all spectra of all minerals. There is no reliable way to physically measure or precisely calculate with modeling, e.g. FEFF¹⁶ where the arctangents are, therefore, this choice is somewhat arbitrary, but it is made consistently across all spectra of all minerals.”

The manual adjustment of the symmetry of peaks 1 and 3 is not described. Is this a

mathematical transformation?

Thank you. We now explained on SI pages 9: “4. All spectra were rigidly shifted until peak 1 was at exactly 352.6 eV, then the maxima of peaks 1 and 3 were adjusted manually to be perfectly symmetric around the maximum if the very tip (1 or 2 data points) of a peak happened to be asymmetrical due to sampling with 0.1 eV step during data acquisition and the physical peak not falling precisely on one of the energies spaced 0.1 eV apart. For example, if a peak is at 352.63 eV but we only sample 352.6 and 352.7 eV, the latter 2 datapoints would not be at the same intensity. In this case we manually adjusted the intensity, keeping the highest intensity point as the center of the peak, and manually moving the intensity of the 2 datapoints before and after it up or down to make the peak as symmetric as possible around the center. This minimal adjustment only affected 2 data points out of the 121 present in each spectrum, and it never shifted the energy or intensity of a peak.”

p. 8 Furthermore, the following aspects of the fitting routine should be verified:

Step 5) It is not clear, what the left and right cursor in the GG macro define

Thank you. We now explained on SI page 9: “5. The intensity of all spectra was adjusted to 0 and 10, pre-edge and peak 1, with left cursor in GG Macros 15 at 345 eV and right cursor at 352.6 eV. The left and right cursors define these two positions: pre-edge at 345 eV, 352.6 eV on top of peak 1.”

Step 6) It is not clear, whether this approach of “liberating” fit parameters one at a time can introduce a bias and can depend on the order of liberating parameters

Thank you. We now explained on SI page 9: “Fig. S3 shows that the order in which peaks are feed does not affect the result of peak fitting.”

Step 7) It seems that the constraint required here would be to assume the peak to be positive to avoid fitting the “dip” which should not require manual “trial and error”

We always enforced all peaks to be positive. In the peak 2 position, however, the fitting misbehaved. The dip between peaks 2 and 3 was present in all synthetic and biogenic minerals, but its depth varied. If we included this dip in the fit it made peak 2 artificially narrow as the software tried to minimize the overlap between peaks 2 and 3. We now clarify this process on SI page 9: “7. Peaks that did not fit well were isolated and fit only within the energy range of that peak, excluding overlap with neighboring peaks. This was always necessary for peak 2 because of a significant intensity dip between peaks 2 and 3 in all spectra. Isolating peak 2 and not allowing the software to fit it to the dip made it possible to fit the lineshape of peak 2 itself perfectly (zero residue). Some peaks required fitting by trial and error while holding parameters, not letting them freely vary, to force the software to recognize important sections of the lineshape. This was necessary in energy ranges with heavily overlapping peaks such as the aragonite peaks 2 and 2' or peaks 4, 4', and 4'', as shown in Table S4, to avoid the fitting peaks in non-physical

ways. For instance, during fitting with completely free peaks 2 and 2', peak 2 was fit with a reasonable amplitude and width, and matched the data reasonably well, whereas peak 2' had enormous width and/or very small amplitude, which helps the fit outside the region of peak 2' but not on peak 2', where the fit and the data did not match at all."

Step 9) It seems that the same result could have been achieved by constraining the range of the peak position

Thank you. We respectfully disagree. The peak energy position and range is not relevant here, what matters is the amplitude. This was confusing and is now much clearer after the addition of the new figure, so we wrote on SI page 9: "This is clearly shown in the CCHH peaks 4', 4", and 4'" in Fig. S3."

p. 8/9 Whereas the project itself is very honourable and certainly merits a broad public propagation, I believe that the description of the methodology of the data analysis is not the right place. The description should be limited to a more factual account.

We removed any mention of the Cnidarians.

There is no guarantee that a converging solution is always the right solution to a problem, especially as the participants involved in the data analysis are not necessarily working on their solutions fully independently.

We agree with the reviewer that our previous statement was somewhat exaggerated, we have now better explained the role of convergent solutions in our data processing on SI page 12: "This processing was repeated on the same set of data by multiple people in parallel and multiple times by the same person, assessed both quantitatively and qualitatively. Quantitatively including χ^2 values and percent error in identifying known phases, and qualitatively for example observing if the metastable phases appeared in places where they make sense and a visual inspection of the fit quality. With each repetition, each decision leading to improvement was held constant, eventually leading to the data processing presented here."

For the scope of unbiased data analysis, an anonymous peer-based analysis seems a more robust approach.

While we agree with the reviewer conceptually, the complexity of data processing and interpretation makes this type of analysis next to impossible. It takes months to get a single user proficient enough with all software involved to be able to participate in the analysis. As there are functionally no other places to learn how to process our data, it is simply not possible to get a group of people together that all know how to process the data and are anonymous from one another.

We have removed previous statements about this being the most robust method in order to leave room for improvement, but as of right now the suggested method is not

possible.

The quality of the fit should additionally be compared using a reduced χ^2 to take the difference in degrees of freedom between a 3-, 4- or 5-component fit into account.

We thank the reviewer for pointing out this method of statistical comparison and have integrated it into **Table S3** (formerly **S2**). We have simply replaced the χ^2 with the reduced χ^2 because the reduced χ^2 contains all of the same information while correcting for the increased degrees of freedom.

In terms of the 12 decisions that are to be made by the Cnidarians, some of them might be more objective if aided by computation. This is true for all criteria that are quantifiable, such as the alignment of images, the definition of noise level and the appropriate selection of the spectral region to be analyzed.

We thank the reviewer for identifying some places where we had not elaborated on how computation was used to aid the decision making process. We have added two explanations to the SI:

SI page 11: “1. when a stack of images is well aligned (e.g., within $\frac{1}{2}$ pixel for lateral motion), and what method to use in aligning a misaligned stack. Alignment is judged in the PEEMVision software package, which allows the user to view the stack of images as a movie and watch for drift of features. The user then identifies a distinct feature that stays constant across all images, and uses one of various alignment algorithms (align, align vertical, align horizontal, align 4x oversample, etc.) to align that feature across all images.”

SI page 12: “This processing was repeated on the same set of data by multiple people in parallel and multiple times by the same person, assessed both quantitatively and qualitatively. Quantitatively including χ^2 values and percent error in identifying known phases, and qualitatively for example observing if the metastable phases appeared in places where they make sense and a visual inspection of the fit quality. With each repetition, each decision leading to improvement was held constant, eventually leading to the data processing presented here.”

The description of the analysis process is very technical and lacks a description of the underlying mathematical process (e.g. what do the cursors do).

We thank the reviewer for pointing out this point of confusion, we have now clarified in the **SI page 11:** “The cursors define on-peak and off-peak energy positions, then Igor subtracts the off-peak images from on-peak images, to obtain difference maps.”

p. 11/12 The description of the SINS-FTIR approach is lacking some detail, especially concerning the tip and substrate used, the sample thickness and the experimental setup itself (commercial device?) The authors should look into the penetration depth of the IR signal. The limitation to the penetration depth of 20 nm in all dimensions is not accurate.

Whereas the lateral resolution is dominated by the tip size, the penetration depth essentially depends on the tapping amplitude of the cantilever as well as the harmonic that is chosen in the analysis. There are studies dedicated to obtaining some kind of tomography using SINS-FTIR. Furthermore, there is no consensus in literature concerning the penetration depth under standard conditions, with values reported varying from tens of nm to a couple of hundred nm.

We thank the reviewer for suggesting we add more information. We now wrote on **SI pages 14-15**: “SINS-FTIR, on the other hand, is more surface sensitive than ATR-FTIR. Its sensitivity decays rapidly from the surface, thus its probing depth is typically on the order of the tip radius, in this case ~20 nm. Subsurface sampling is possible, depending on the material, tapping amplitude, and other experimental parameters, but it does not dominate the signal. The higher surface sensitivity and the higher lateral spatial resolution increase the ability of SINS-FTIR to differentiate heterogeneous surface phases from bulk aragonite.

SINS-FTIR data were collected at the Advanced Light Source, Lawrence Berkeley National Lab over multiple sessions using both the Neaspec IR neaSCOPE instrument at Beamline 2.4 and a custom modified Bruker Innova AFM at Beamline 5.4. SINS is described in detail in previous work^{17, 18}. Briefly, broadband IR synchrotron radiation is introduced into an asymmetric Michelson interferometer in which one half of the incident light is focused onto a platinum-coated silicon AFM tip, oscillating in close proximity to the sample, and the other half is reflected off a moving flat mirror. The light scattered by the nearfield interaction of the tip with the sample interferes with the light reflected from the moving mirror and is detected by a mercury cadmium telluride (MCT) infrared detector. The signal is demodulated at higher harmonics of the tip-tapping frequency with a lock-in amplifier to remove far-field background contributions and then Fourier transformed to yield both an amplitude and a phase spectrum, which approximate the reflectance and absorbance spectrum, respectively¹⁸. Here, we display only 2nd harmonic amplitude data because the phase values were undefined in regions where the amplitude signal was near zero.”

We also wrote on **SI pages 15**: “SINS spectra of reference materials were obtained from powder samples, pressed onto carbon tape with a KBr press to provide a flat surface for AFM imaging (see Fig. S8). Coral samples were embedded in epoxy and then polished. Polydimethylsiloxane (PDMS) often contaminates the AFM tips used for SINS. The amount of PDMS contamination varied from tip to tip, causing some spectra to have stronger PDMS peaks than others. The tip used during acquisition of the ACCH₂O, ACC, and aragonite spectra had a larger PDMS contamination, causing distinctive peaks especially around 825 cm⁻¹ and 1270 cm⁻¹. All CCHH and coral v₃ peaks were at different frequencies compared to PDMS peaks; thus, they were not affected by PDMS contamination. The peaks associated with PDMS are labeled by pink lines in Fig. 4.

We note that other IR nearfield AFM-based techniques besides SINS that have been used to probe calcium carbonate biominerals, including Photo-induced Force Microscopy (PiFM).^{19, 20} “

Finally, some issues have been identified in the extended data section of the manuscript, which the authors should address:

S1) There is a huge difference between the number of single-pixel spectra recorded for e.g. 1-ACC (14) and 2-CCHH (20402). Is this difference taken into account when considering the single components for the fit? Presumably the spectrum of 1-ACC is much noisier than the one for 2-CCHH

We thank the reviewer for pointing out the discrepancy in the number of spectra between certain minerals, as it led to the discovery of an error in the previous manuscript. Miscommunication led to the incorrect low number of ACC and MHC spectra, which were acquired by using “binned” spectra. Binned spectra are simultaneously extracted average spectra from a defined area, and thus each binned spectrum contains many single-pixel spectra. The correct number of actual single-pixel spectra are now reported for each component in Table S1, and the numbers are much more similar to one another, ranging from 3200 to 20402. SI page 16.

S2) The improvement in the description when replacing the vaterite component with MHC or CCHH raises the question, whether no vaterite is present or whether the model fails to distinguish between the two phases.

We agree with the reviewer that the previous 4cmp analysis was inconclusive on vaterite (and yes, the vaterite spectrum is completely different! See Fig. 1), which led us to re-process the entire data set, one more time, using a more physically meaningful energy range, described on SI page 11: “3. within which energy range the unknown spectrum in each pixel should be fit by a linear combination of the selected component spectra. After many trials, we converged on the energy range 340-355 eV. This includes all of the pre-edge, and all of the Ca peaks, but it excludes the post-edge, because the intensity of the featureless post-edge is directly proportional to Ca concentration in each pixel, whereas the lineshape in the energy region joining the highest energy peak 1 and the flat post-edge (354-355 eV) is strongly mineral-dependent and should therefore be included in mineral assignment analysis.”

This change was not requested by the reviewer, but it made the analysis much better. Combined with the previously suggested reduced χ^2 , we show that the use of vaterite does not significantly improve the fit, even in pixels determined to be vaterite, in Table S3 (formerly S2). We hope the reviewer will find these new data more compelling.

S3) According to the table the fit parameter of p3 is 0. Probably the value is smaller than 0.0005 and a more appropriate representation should be chosen for the table.

The 0.000 value was really zero, in fact we were not using a 3rd order polynomial but a 2nd order one. A 3rd order polynomial is what Igor converts to by default in the export of

text data files, after the fit, not during the fit. We have now corrected this mistake in Table S4 (formerly S3), SI page 20, and thank the reviewer for spotting it! Very useful!

As described in the caption, the area under the curve is represented, rather than the amplitude. The table should be adapted accordingly.

We more clearly explained in Table S4 caption that by “amplitude” we mean the area under the peak, not the peak intensity. Again, this is Igor nomenclature not ours, and we are conforming to it so readers that want to reproduce the results with the same software will have an easier time. See SI pages 20-21.

As discussed throughout the text, the main parameter to distinguish the components is the aspect ratio. These values should be included here.

The entire component spectrum was used for each component, not one peak at a time, or ratio of peak intensities. A linear combination of component spectra fitted every pixel spectrum in a stack of images to produce an MM. The words “aspect ratio” indicated that the relative intensities of peaks vary across all minerals and are therefore characteristic of each spectrum. We have now removed the words “aspect ratio” and left “relative intensities” or simply “intensities”, which is probably less confusing. We replaced these on pages 3 and 5.

For some of the peaks of the aragonite, all parameters are fixed. It is not clear how these parameters were obtained.

This process and the reasoning behind it is now better explained on SI page 9: “Some peaks required fitting by trial and error while holding parameters, not letting them freely vary, to force the software to recognize important sections of the lineshape. This was necessary in energy ranges with heavily overlapping peaks such as the aragonite peaks 2 and 2' or peaks 4, 4', and 4'', as shown in Table S4, to avoid the fitting peaks in non-physical ways. For instance, during fitting with completely free peaks 2 and 2', peak 2 was fit with a reasonable amplitude and width, and matched the data reasonably well, whereas peak 2' had enormous width and/or very small amplitude, which helps the fit outside the region of peak 2' but not on peak 2', where the fit and the data did not match at all.”

Fig. S1) The peak position of the 0-ACC-H₂O seems to be almost identical to 2-CCHH according to Table S3 and hence not distinct from all other carbonates.

Thank you for catching this misleading phrase. We now explained in Fig. S1 caption, SI page 24: “Peak 2 for CCHH occurs at 351.5 eV, distinct from all other carbonates except for ACCH₂O. The latter is very different from CCHH, in the peak 4 region: ACCH₂O has only a broad and low shoulder at 348.09 eV, CCHH instead has four distinct peaks, termed 4, 4', 4'', 4''' at very different energy positions and with much smaller peak width. The amplitude of peak 4 at 348.24 is much greater for CCHH than for ACCH₂O, and the amplitude for 4', 4'', 4''' are much smaller than peak 4 for

ACCH₂O. All peak parameters are listed in Table S4.”

Fig. S3) The pixel size is given as 19 nm, better than the best lateral resolution as described in the main text

The pixel size can be larger or smaller than the best resolution. In this case the field of view was 20 μm and there were 1054 pixels horizontally, thus, 19 nm pixels. Indeed, a little oversampled (a little too many pixels in the image) compared to the lateral resolution of the microscope, but not too far.

Fig. S6) The figure contains an extensive caption of almost one page that would better fit into the main text or the description of the methodology.

We agree with the reviewer that the caption was too long, and the SI was not the right place to present these important corroborating data. We moved the SINS-FTIR figure to the main text, incorporated part of the caption into the text, and discussed the data more in detail in the results. See pages 9-11.

The spectra shown presumably correspond to the absorption of the sample. In SINS-FTIR, the infrared absorption spectrum is typically obtained from the phase signal of the measurement. Some information on the computation of the spectra should be added.

We much expanded the data description on SI page 14-15. All of the SINS spectra were spectral amplitude, which came from the real portion of the Fourier transform of the interferogram.

It is not clear why different synthetic spectra are chosen for the comparison in S6A and S6B.

Panels 4A and 4B (formerly S6A and S6B) were meant to display the variability in both coral and synthetic minerals in SINS, as now explained on page 9: “The peak positions and splitting in the ν_3 region vary significantly across the coral surface spectra, which may correspond to multiple levels of hydration. Similar variability was also observed across different grains of synthetic CCHH powder, as shown by a comparison of syn CCHH I and II spectra in A and B.” much more discussion of SINS data is on pages 9-11.

For a better link between the SINS spectra of the sample and the spectra of the synthetic minerals, a classical ATR-FTIR spectrum of the synthetic material should be obtained. This would be helpful for an unambiguous assignment.

We thank the reviewer for the suggestion, but while we agree that a comparison with ATR-FTIR would be useful in concept, we collected ATR-FTIR spectra on each of the synthetic minerals and saw enough of a difference between the ATR-FTIR and SINS spectra in the synthetic standards that they did not help with identification of the coral unknown spectra. Surface and bulk spectra, evidently, are different in the IR as much

they are in the soft-x-ray region! For example, acquiring simultaneously total electron yield (TEY) and x-ray fluorescence (XRF) spectra of the same powder shows completely different spectra, originating from the surface and the bulk, respectively.

The authors observe a significant shift of the peak frequencies in different directions (shift up to 50 cm⁻¹ downwards and up to 65 cm⁻¹ upwards). There is no explanation for this shift and in particular for the different shift for the sample described as “coral surface III” given.

We thank the reviewer for pointing out the lack of explanation of the SINS-FTIR data. We now added to the main text a list of potential causes for these peak shifts, on page 9: “These shifts in the ν_3 peaks of biogenic CCHH and MHC and change in peak splitting could be due to multiple factors, including mixed signal from aragonite, different hydration levels relative to their synthetic counterparts, interactions with organic molecules trapped within the biomineral, or trace elements in the biominerals, any of which could potentially cause a change in the asymmetric stretching of the carbon-oxygen bonds in carbonate ions²⁷.”

Fig S9) The representation of experimental spectra of the sample interspersed with spectra of the synthetic material is not very clear. The synthetic spectra should be highlighted (e.g. by dotted lines)

We thank the reviewer for this suggestion. We have outlined the Cni16 component spectra in Fig. S11 (formerly S9) on SI page 34.

Reviewers' Comments:

Reviewer #1:

Remarks to the Author:

In the revised version of the text, all my doubts and suggestions were properly addressed.

Reviewer #2:

Remarks to the Author:

The authors have extensively revised and significantly strengthened their manuscript. Its overall presentation, wording, and structure have significantly improved. I welcome the changes made to the supplementary especially in adding more details to the sample preparation and analytical methods and appreciate the more thorough discussion on potential phase transformation pathways in the main text. The choice to include two scale bars in some of the figures may appear a bit unusual but I believe they are a valid approach to convey both pixel and biomineral feature sizes.

I have very few remaining remarks that I would like the authors to consider before recommending publication in Nature Communications:

(1) Page 16 line 14: "One key advantage of crystalline vs. amorphous precursors is (...)" I recommend rephrasing to "One analytical key advantage..." to emphasize to the reader that this sentence is about the detectability of the phases.

(2) Page 16, line 21: "Photo-induced Force Microscopy (PiFM) with 5 nm lateral resolution and 20 nm depth sensitivity" The given depth resolution is not correct, while it's true that its exact depth is dependent on many factors in the analytical setup as well as sample properties it is about 5 nm in sideband modulation (the most commonly used mode) as it is demonstrated to have monolayer sensitivity. A deeper signal generation can be achieved when using direct mode but this is less commonly utilized. In the referenced publication from Otter et al., 2021, the depth penetration depth is given as between 2 to 30 nm (depending on the selected mode). The mechanisms behind the signal depth are explained in more detail in NNowak, D., et al. (2016): Nanoscale chemical imaging by photoinduced force microscopy. Science advances, 2(3). Hence, I suggest changing "20 nm" to "about 5 nm for sideband mode" (this applies also to the Supplementary).

(3) The peak splitting observed and attributed to CCHH has been discussed to result from several factors that are all related to atomic order structuring of the C-O. However, in AFM-IR methods peak splitting can also result as a technical artifact resulting when the cantilever frequencies when the excitation source and cantilever frequencies are out of sync and the intensities exceed the ideal range. I believe this type of peak splitting would look different and hence believe the SINS data to be robust (i.e., indeed resulting from the atomic structure of the materials). However, I think the authors should investigate and consider if there are any potential technical artifacts that may lead to peak splitting and make sure to rule it out in their data.

In summary, I am very excited about all the choices made to revise across all areas of the manuscript and supplementary and am keen to see this published. Well done authors!

Reviewer #3:

Remarks to the Author:

The revised version of the manuscript has been significantly improved and both the way the data is presented, and the way the experimental and analytical procedures are described make it much easier now to understand and reproduce the important results of this study.

I would like to thank the authors for their detailed feedback to my comments and the extensive improvement of the manuscript.

I would only like to suggest two minor things for the SINS-FTIR section:

- The authors should include the tip used for the experiment and the corresponding resonance frequency.
- Whereas the authors describe in detail the reasons for the observation of various peak-shifts in the SINS-FTIR data in terms of local sample variability, potential artifacts due to far-field effects that can commonly be found for lower harmonics (such as O₂) should briefly be mentioned as well.

I would like to recommend the manuscript for publication in its present form considering the minor changes mentioned before.

REVIEWER COMMENTS

Reviewer #1 (Remarks to the Author):

In the revised version of the text, all my doubts and suggestions were properly addressed.

Thank you!

Reviewer #2 (Remarks to the Author):

The authors have extensively revised and significantly strengthened their manuscript. Its overall presentation, wording, and structure have significantly improved. I welcome the changes made to the supplementary especially in adding more details to the sample preparation and analytical methods and appreciate the more thorough discussion on potential phase transformation pathways in the main text. The choice to include two scale bars in some of the figures may appear a bit unusual but I believe they are a valid approach to convey both pixel and biomineral feature sizes.

Thank you!

I have very few remaining remarks that I would like the authors to consider before recommending publication in Nature Communications:

(1) Page 16 line 14: “One key advantage of crystalline vs. amorphous precursors is (...)” I recommend rephrasing to “One analytical key advantage... “ to emphasize to the reader that this sentence is about the detectability of the phases.

On page 16, line 14 now says “One **analytical** key advantage of crystalline vs. amorphous precursors is that future studies can explore their presence....”

(2) Page 16, line 21: “Photo-induced Force Microscopy (PiFM) with 5 nm lateral resolution and 20 nm depth sensitivity” The given depth resolution is not correct, while it's true that its exact depth is dependent on many factors in the analytical setup as well as sample properties it is about 5 nm in sideband modulation (the most commonly used mode) as it is demonstrated to have monolayer sensitivity. A deeper signal generation can be achieved when using direct mode but this is less commonly utilized. In the referenced publication from Otter et al., 2021, the depth penetration depth is given as between 2 to 30 nm (depending on the selected mode). The mechanisms behind the signal depth are explained in more detail in NNowak, D., et al. (2016): Nanoscale chemical imaging by photoinduced force microscopy. Science advances, 2(3). Hence, I suggest changing “20 nm” to “about 5 nm for sideband mode” (this applies also to the Supplementary).

On page 16 lines 20-22 now say: “Photo-induced Force Microscopy (PiFM), with 5 nm lateral resolution and about 5 nm depth sensitivity in sideband mode, is also sensitive to carbonate FTIR spectroscopy⁵⁵.”

(3) The peak splitting observed and attributed to CCHH has been discussed to result from several factors that are all related to atomic order structuring of the C-O. However, in AFM-IR methods peak splitting can also result as a technical artifact resulting when the cantilever frequencies when the excitation source and cantilever frequencies are out of sync and the intensities exceed the ideal range. I believe this type of peak splitting would look different and hence believe the SINS data to be robust (i.e., indeed resulting from the atomic structure of the materials). However, I think the authors should investigate and consider if there are any potential technical artifacts that may lead to peak splitting and make sure to rule it out in their data.

We added a comment to this effect in the SI. See SI page 14: “We observed peak splitting in CCHH and MHC, both synthetic and biogenic. We did not observe such splitting in any other regions of the skeleton other than near the forming surface. Peak splitting can, in principle, result from a technical artifact, observed when the AFM cantilever frequencies and the excitation source are out of sync and the intensities exceed the ideal range. Multiple lines of evidence suggest that the observed peak splitting is indeed characteristic of CCHH and MHC, and not an artifact. First, we observe precisely the same peak splitting in biogenic and synthetic CCHH and MHC. Second, we never observed it in synthetic ACC or aragonite, only in synthetic CCHH and MHC crystals. Third, in corals, we only observed peak splitting near the forming surface of coral skeletons. The observed peak splitting, therefore, cannot be artifact.”

In summary, I am very excited about all the choices made to revise across all areas of the manuscript and supplementary and am keen to see this published. Well done authors!

Reviewer #3 (Remarks to the Author):

The revised version of the manuscript has been significantly improved and both the way the data is presented, and the way the experimental and analytical procedures are described make it much easier now to understand and reproduce the important results of this study.

I would like to thank the authors for their detailed feedback to my comments and the extensive improvement of the manuscript.

Thank YOU!

I would only like to suggest two minor things for the SINS-FTIR section:

- The authors should include the tip used for the experiment and the corresponding resonance frequency.

We already had already said that we used a platinum-coated silicon AFM tip. See SI, page 13, "...light is focused onto a platinum-coated silicon AFM tip...".

We added this on SI 14: "The tapping oscillation frequency was either 235 kHz, 241 kHz, or 252 kHz."

- Whereas the authors describe in detail the reasons for the observation of various peak-shifts in the SINS-FTIR data in terms of local sample variability, potential artifacts due to far-field effects that can commonly be found for lower harmonics (such as O2) should briefly be mentioned as well.

We added a comment about far-field artifacts on SI page 14: "Other artifacts, including far-field effects and tip-sample coupling, can cause peak shifts. Far-field signals can be distinguished from near-field signals better at higher harmonics but at the cost of worse signal to noise ratio, thus, Bechtel et al. concluded that using second harmonic, as done in all of this work, was the best compromise¹⁸. Far-field effects, however, are expected to affect vast regions of the sample, not isolated 20nm pixels, which is where CCHH and MHC were observed in this work."

I would like to recommend the manuscript for publication in its present form considering the minor changes mentioned before.

Thank you!

Reviewers' Comments:

Reviewer #2:

Remarks to the Author:

All my comments have been addressed and I am very happy with the revised version of the manuscript. I have no further comments and suggest publication in Nature Communications.

Reviewer #3:

Remarks to the Author:

I would like to thank the authors for the revision of the manuscript. All remarks have been considered.

I would like to recommend the manuscript for publication in its present form.